# Analysis of the Equilibrium Distribution of Ligands in Heterogeneous Media–Approaches and Pitfalls

**DOI:** 10.3390/ijms23179757

**Published:** 2022-08-28

**Authors:** Maria João Moreno, Luís M. S. Loura, Jorge Martins, Armindo Salvador, Adrian Velazquez-Campoy

**Affiliations:** 1Coimbra Chemistry Center—Institute of Molecular Sciences (CQC-IMS), University of Coimbra, 3004-535 Coimbra, Portugal; 2Department of Chemistry, Faculty of Sciences and Technology, University of Coimbra, 3004-535 Coimbra, Portugal; 3Faculty of Pharmacy, University of Coimbra, 3000-548 Coimbra, Portugal; 4Centro de Ciências do Mar (CCMAR/CIMAR, LA) and DCBB-FCT, Universidade do Algarve, Campus de Gambelas, 8005-139 Faro, Portugal; 5CNC—Center for Neuroscience and Cell Biology, 3004-504 Coimbra, Portugal; 6Institute for Interdisciplinary Research, University of Coimbra, 3030-789 Coimbra, Portugal; 7Institute for Biocomputation and Physics of Complex Systems (BIFI), Joint Unit GBsC-CSIC-BIFI, Universidad de Zaragoza, 50018 Zaragoza, Spain; 8Departamento de Bioquímica y Biología Molecular y Celular, Universidad de Zaragoza, 50009 Zaragoza, Spain; 9Instituto de Investigación Sanitaria de Aragón (IIS Aragon), 50009 Zaragoza, Spain; 10Centro de Investigación Biomédica en Red en el Área Temática de Enfermedades Hepáticas Digestivas (CIBERehd), 28029 Madrid, Spain

**Keywords:** binding affinity, partition coefficient, proteins, lipid bilayers, biomembranes, Stern–Volmer plots, Scatchard plots

## Abstract

The equilibrium distribution of small molecules (ligands) between binding agents in heterogeneous media is an important property that determines their activity. Heterogeneous systems containing proteins and lipid membranes are particularly relevant due to their prevalence in biological systems, and their importance to ligand distribution, which, in turn, is crucial to ligand’s availability and biological activity. In this work, we review several approaches and formalisms for the analysis of the equilibrium distribution of ligands in the presence of proteins, lipid membranes, or both. Special attention is given to common pitfalls in the analysis, with the establishment of the validity limits for the distinct approaches. Due to its widespread use, special attention is given to the characterization of ligand binding through the analysis of Stern–Volmer plots of protein fluorescence quenching. Systems of increasing complexity are considered, from proteins with single to multiple binding sites, from ligands interacting with proteins only to biomembranes containing lipid bilayers and membrane proteins. A new formalism is proposed, in which ligand binding is treated as a partition process, while considering the saturation of protein binding sites. This formalism is particularly useful for the characterization of interaction with membrane proteins.

## 1. Introduction

Biological systems are very heterogeneous media, with macromolecules or supramolecular aggregates dispersed in an aqueous medium. The hydrophobic effect is the major force that drives the formation of these structures [1,2] where the dispersed structures are characterized by a polar or ionic shell and a nonpolar core. Small molecules distribute between the aqueous phase and the non-aqueous structures as a result of the established interactions. The hydrophobic effect is the major force for non-specific binding, while polar interactions (type, configuration and geometry) are responsible for interaction specificity. The distribution of the small molecules (ligands) between the distinct structures determines their activity and availability.

The most common examples of heterogeneous media are proteins in aqueous solution [3]. Most of these macromolecules fold into very organized and well-defined structures, with distinct regions where the other molecules may bind with higher or lower affinity and specificity [4,5].

Biological membranes are another example of heterogeneous media, formed by amphiphilic lipids assembled in a bilayer and containing proteins embedded in the bilayer or associated with its surface [6,7]. A major difference between lipid membranes and proteins is the inexistence of well-defined binding sites in the former. Being formed from a large number of molecules associated by many weak forces, lipid bilayers of biomembranes are generally in the fluid phase and can dissolve or incorporate small molecules without changing their properties, namely their ability to dissolve additional molecules. Thus, although the properties of the membranes vary strongly across the bilayer, the distribution of small molecules between the aqueous medium and membranes is usually well-defined by a partition coefficient or a partition constant [8,9,10].

Other examples of heterogeneous media are micelle solutions, with properties differing from the previous two examples. Similar to lipid bilayers, micelles are also formed by the association of many amphiphilic molecules held together by weak forces. They are therefore dynamic structures, able to dissolve small molecules. However, due to their small size (comparable to that of proteins), micelles saturate at relatively small numbers of solute occupancy, with the average number of solutes per micelle <*n*> being usually ≤1 [11]. At small solute concentrations, the distribution between the aqueous medium and the micelles is usually well described by a partition. However, similarly to lipid bilayers, the local concentration of solute is not homogeneous within the micelle [12,13]. For <*n*> = 1, the number of solutes encountered per micelle usually follows a Poisson distribution, with about 40% of the micelles being without solute, while 26% of the micelles having 2 or more solutes. The distinct and small number of solutes in the micelles leads to stochastic effects, strongly influencing the system’s properties and reactivity [14,15,16,17].

Another distinctive characteristic of heterogeneous media is their dynamics. Folded proteins are structures formed by a single macromolecule or a few polypeptide chains with relatively strong interactions between them. Conversely, lipid bilayers and micelles are supramolecular structures held together by weak forces. The number of amphiphiles in each micelle varies from a few molecules (e.g., bile salt micelles [18]) to a few hundred [19,20], while biomembranes are formed by millions of amphiphilic lipids. The amphiphiles that associate as micelles also have a higher hydrophilic/lipophilic balance and are therefore more soluble in the aqueous medium as monomers [21]. Both properties lead to faster dynamics of amphiphile insertion/desorption and assembly/disassembly of the supramolecular structure for the case of micelles, with important consequences for solute distribution [22,23,24,25]. Given the particularities of micellar heterogeneous media, and their relatively lower importance in biological systems, they will not be further addressed in this work.

When considering the distribution of small molecules in heterogeneous media, it is important to distinguish between affinity and specificity. The binding affinity of a ligand to a protein or biomembrane reflects to a high extent its poor affinity for the solvent (its hydrophobicity). In turn, binding specificity depends on the establishment of several interactions with the binding agent in a well-defined, three-dimensional arrangement of the groups involved in the interaction. Although the maximization of affinity usually leads also to a high specificity, ref. [26] the latter may vary widely for the case of ligands with low-to-moderate hydrophobicity [27]. Interactions with lipid bilayers are usually not specific, with the association affinities depending essentially on the fluidity and charge of the membrane [9,28,29,30,31,32,33,34,35,36,37,38]. Some exceptions may occur for the case of membranes with lateral heterogeneity and/or when glycolipids are present [39,40,41,42]. In contrast, proteins have well-defined binding sites with a characteristic spatial distribution of functional groups that lead to a stronger interaction with some specific molecules [43,44,45,46,47]. However, for some proteins a wide variability is observed in the properties of ligands with similar affinity (with serum albumins being a striking example), showing that binding to proteins can also be non-specific. This review only addresses the methodologies and formalisms for characterization of binding affinity.

Because proteins have well-defined binding sites, the characterization of ligand affinity may be done following two approaches: (i) from the fraction of protein with ligand bound; and (ii) from the fraction of ligand bound to the protein. Methodologies based on both approaches will be addressed in this review, and in principle both lead to the same parameters. In contrast, due to the absence of well-defined binding sites in lipid membranes, very different results are obtained depending on whether the fraction of ligand bound (increasing the amount of membrane) or the local concentration of ligand in the membrane (increasing the amount of ligand) is followed. The former approach provides the ligand binding affinity for an unperturbed bilayer, while the later provides information on the saturation of the membrane with the ligand (which corresponds necessarily to a membrane with altered properties).

It is the objective of this review to compile different formalisms and experimental approaches for the characterization of the binding affinity of small molecules to proteins and lipid membranes. Focus is given to practical aspects, with the clear identification of the applicability of different formalisms and common pitfalls. The complexity of the considered heterogeneous systems increases throughout the review, from proteins with a single binding site to biomembranes containing both proteins and lipid bilayers.

## 2. Proteins in Solution

### 2.1. Proteins with a Single Binding Site

#### 2.1.1. Analytical Solution of the Full Equation

Ligand association is more frequently followed through variations in a protein property that is proportional to the advance of the binding process (i.e., saturation fraction). Several properties may be followed, including the protein enzymatic activity, ref. [48] protein structure or intrinsic fluorescence, refs. [49,50,51,52,53] changes in the charge or mass, refs. [54,55,56] or properties that reflect the binding itself, such as the heat evolved or changes in the ionization of the ligand or protein [47,49,56].

If the protein contains only one type of binding sites (one or several equivalent and independent binding sites on each protein molecule), ligand association may be described by the following scheme and equilibrium relations:(1)$$\begin{array}{l} L_{W}+B\overset{K_{b}}{↔}L_{B}\;;\;K_{b}=\frac{\left[L_{B}\right]}{\left[L_{W}\right]\left[B\right]};\;K_{d}=\frac{1}{K_{b}}\\ \left[L_{B}\right]=K_{b}\left[L_{W}\right]\left[B\right];\;\left[L_{B}\right]=\frac{K_{b}\left[L_{W}\right]\left[B_{T}\right]}{1+K_{b}\left[L_{W}\right]} \end{array}$$
where $\left[L_{B}\right],\;\left[L_{W}\right],\;\left[B\right],\;\mathrm{and}\;\left[B_{T}\right]$ are, respectively, the concentrations of the ligand bound to the protein, the ligand in the aqueous phase, unoccupied binding sites in the proteins, and the total concentration of binding sites (all with respect to the total volume of the solution), whereas $K_{b}$ and $K_{d}$ are, respectively, the equilibrium binding and dissociation constants. Considering the mass conservation equations for this system $\left(\left[L_{T}\right]=\left[L_{W}\right]+\left[L_{B}\right];\;\left[B_{T}\right]=\left[B\right]+\left[L_{B}\right]\right)$ the equation from which the concentration of occupied binding sites may be calculated is:(2)$${\left(x\right)}^{2}-x\left(K_{d}+\left[B_{T}\right]+\left[L_{T}\right]\right)+\left[L_{T}\right]\left[B_{T}\right]=0;\;x=\left[L_{B}\right]$$
where the dissociation is used instead of the association constant due to the simplification of the resulting equation.

To use this quadratic equation for the quantitative analysis of experimental results, it is first necessary to identify which of the two mathematically possible solutions, Equation (3), has physical meaning.
(3)$$\begin{array}{l} x_{+}=\left[L_{B}\right]=\frac{\left(K_{d}+\left[B_{T}\right]+\left[L_{T}\right]\right)+\sqrt{{\left(K_{d}+\left[B_{T}\right]+\left[L_{T}\right]\right)}^{2}-4\left[L_{T}\right]\left[B_{T}\right]}}{2}\\ x_{-}=\left[L_{B}\right]=\frac{\left(K_{d}+\left[B_{T}\right]+\left[L_{T}\right]\right)-\sqrt{{\left(K_{d}+\left[B_{T}\right]+\left[L_{T}\right]\right)}^{2}-4\left[L_{T}\right]\left[B_{T}\right]}}{2} \end{array}$$

In the case of association between two entities, it may be easily identified that the physically meaningful solution is always $x_{-}$, because $x_{+}$ is higher than $\left[B_{T}\right]$ and/or $\left[L_{T}\right].$ Where not evident, the physically meaningful solution may be decided by calculating $x_{\pm }$ over a large range of parameter and concentration values, with the constraints that it must be >0, <$\left[B_{T}\right]$ and <$\left[L_{T}\right]$. When the protein saturation is experimentally followed, the concentration of ligand is the independent variable, and the association affinity $\left(K_{b}\;\mathrm{or}\;K_{d}\right)$ may be found through the best fit of the correct solution of the quadratic equation to the experimental results. The solution of Equation (3) is valid in all conditions, low- or high-affinity binding, comparable or dissimilar concentrations of ligand and protein.

Figure 1 shows the dependence of the protein saturation, with $\left[L_{T}\right]$ (top plots) and with the $\left[L_{T}\right]/\left[B_{T}\right]$ ratio (bottom plots) for several values of the association affinity. As the affinity of the ligand to the protein increases (higher $K_{b}$ or lower $K_{d}$), the protein becomes saturated at lower ligand concentrations and at lower ratios of total ligand to protein-binding sites. The representation of the x axis in the logarithmic scale (plot B and D), which is proportional to the natural thermodynamic variable (the chemical potential), facilitates the interpretation of the results. For high-affinity binding, the protein is fully saturated for $\left[L_{T}\right]/\left[B_{T}\right]$ > 1, smoothly decreasing to zero at lower ligand concentrations. The decrease in the binding affinity leads to a lower fraction of protein saturation at the same $\left[L_{T}\right]$ (or ratio $\left[L_{T}\right]/\left[B_{T}\right]$), with a smoother transition between low and high protein saturation.

Note that the asymmetry of the isotherms obtained for high-affinity binding is due to the representation as a function of $\left[L_{T}\right]$. If the concentration of free ligand is used instead, symmetric isotherms are always obtained, as shown in Appendix A. However, the free ligand concentration is not a known parameter, as it depends on the unknown binding affinity. Thus, to evaluate the adequacy of the conditions regarding the characterization of the binding affinity, one has to consider the known total ligand concentration. From Figure 1 one notes that the binding isotherms for the two highest affinities considered (red and black) are mostly overlapped. This shows that for the total concentration of protein considered in those simulations (10 µM), the highest affinity that may be accurately characterized is $K_{b}$ = 10^7^ M^−1^ ($K_{d}$ = 0.1 µM). For the characterization of higher affinity binding, a lower protein concentration must be used. Illustrating this point, simulations for 1 µM total protein concentration (Appendix A) show that the isotherms for $K_{d}$ = 0.1 µM and 0.01 µM no longer fully overlap in these conditions, allowing their experimental discrimination. The upper limit for accurate estimation of the binding affinity is now $K_{d}$ = 0.01 µM (note the overlap of black and grey symbols in Appendix A).

In the previous figures it was considered that protein saturation was being experimentally followed. If a property of the ligand is monitored instead (e.g., ligand fluorescence), refs. [29,57,58,59,60] the dependent variable is now the fraction of ligand bound, and the total concentration of protein is the independent variable. The concentration of ligand bound is again calculated from the $x_{-}$ solution of Equation (3), from which its fraction may be calculated.

The fraction of bound ligand for several values of the association affinity is shown in Figure 2, as a function of the concentration of binding sites. As expected, the curves are equivalent to those shown in Figure 1 when saturation of the protein is being followed. In fact, the variations observed in the experimentally followed variable (ΔS) are proportional to $\left[L_{B}\right]$, which is the same for a given set of $K_{d}$, $\left[B_{T}\right]$ and $\left[L_{T}\right]$, Equation (4). The normalization of ΔS by its limit value ($\Delta S_{\max}$ corresponding to full protein saturation) leads exactly to the same curve shape as when a property of the protein or ligand is being followed, Equation (5).
(4)$$\left[L_{B}\right]=K_{b}\left[L_{W}\right]\left[B\right]=\frac{\left[L_{W}\right]\left[B\right]}{K_{d}}$$
(5)$$\begin{array}{l} \Delta S=f\left[L_{B}\right]\;;\;\Delta S_{\max}=f{\left[L_{B}\right]}_{\max}\;\\ \Delta S=\frac{\Delta S_{\max}}{{\left[L_{B}\right]}_{\max}}\;\left[L_{B}\right] \end{array}$$

#### 2.1.2. Approximations Commonly Used

Although the analysis methods currently available allow the easy use of the full equation, the use of simplified equations to quantitatively characterize the association between ligands and proteins is still frequent. The two most common approximations are presented below, with a critical analysis of their validity.

When protein saturation is followed, it is commonly assumed that the ligand is in large excess $\left(\left[L_{T}\right]\gg \left[B_{T}\right]\right)$. Under those conditions (corresponding to negligible ligand depletion due to binding), Equation (3) simplifies to the classical hyperbolic equation:(6)$$\frac{\left[L_{B}\right]}{\left[B_{T}\right]}=\frac{K_{b}\left[L_{W}\right]}{1+K_{b}\left[L_{W}\right]}≅\frac{K_{b}\left[L_{T}\right]}{1+K_{b}\left[L_{T}\right]}=\frac{\left[L_{T}\right]}{K_{d}+\left[L_{T}\right]}\;\;\mathrm{for}\;\;\left[L_{T}\right]\gg \left[B_{T}\right]$$

This approximation of excess ligand was originally considered in the context of enzyme kinetics [61], where for in vitro studies the approximation is usually valid because the enzyme is in very low concentrations. However, as shown below, when studying the association between ligands and proteins this approximation may easily become inapplicable and lead to incorrect estimations of the binding affinity.

When binding is followed through a ligand property, it is the fraction of bound ligand that is accessed. The total concentration of protein is usually the independent variable that is varied until nearly all ligand is bound. If it may be considered that the protein is in large excess ($\left[B_{T}\right]\gg \left[L_{T}\right]$), the general equation simplifies to:(7)$$\frac{\left[L_{B}\right]}{\left[L_{T}\right]}≅\frac{K_{b}\left[B_{T}\right]}{1+K_{b}\left[B_{T}\right]}=\frac{\left[B_{T}\right]}{K_{d}+\left[B_{T}\right]}\;\;\mathrm{for}\;\;\left[B_{T}\right]\gg \left[L_{T}\right]$$

Figure 3 (plot A) represents the real saturation of the protein (calculated from Equation (3), circles), and that predicted when assuming excess ligand (continuous lines, Equation (6)). The concentration of binding sites was always 10 µM, and ligand concentration was varied between 10^−1^ and 10^4^ µM. For the lower affinity considered ($K_{d}$ =100 µM), half saturation occurs at a ligand concentration close to 100 µM, where the condition of excess ligand is a good approximation. The best fit of Equation (6) leads to $K_{d}$ = 105 µM, differing by only 5% from the true value. However, as expected, higher affinities lead to significant binding at ligand concentrations comparable to or smaller than the concentration of binding sites. Under those conditions, the approximation of excess ligand is not valid, and the best fit of Equation (6) leads to incorrect parameter estimations. For $K_{d}$ = 10 µM this parameter is overestimated by 50%, and the approximation is totally inappropriate for lower dissociation constants (higher affinities). This inadequacy results from the fact that the most relevant part of the saturation curve for parameter estimation occurs for ligand concentrations that are lower than the concentration of binding sites, violating the assumption of excess ligand. An equivalent behaviour is obtained when the approximation of excess protein is used (Appendix A). In this case, the concentration of binding sites is the independent variable and the fractional binding of ligand is the dependent one.

It should be noted that the general equations for ligand binding to proteins with a single type of binding sites, Equation (1), are valid at all ligand and protein concentrations, for low- and high-affinity binding. The limits of validity of Equations (6) and (7) arise from the substitution of the concentration of free species (ligand or binding sites) with their total concentration. The inadequacy of the approximations of excess ligand or protein is well-known and has been repeatedly highlighted in the literature [61,62]. However, although more careful researchers and specialized software take this into consideration, refs. [47,63,64,65,66] it is still frequently overlooked.

### 2.2. Proteins with More than One Binding Site

#### 2.2.1. Formalism—Microscopic and Macroscopic Association Constants

A commonly encountered difficulty when studying protein/ligand association occurs when the number of binding sites per protein is not well-characterized. If protein saturation may be achieved and followed quantitatively, the macroscopic binding associations may be obtained and from their dependence on protein saturation some conclusions may be drawn regarding the number of binding sites, their similarity and independence. In some conditions, one may also obtain information regarding the microscopic binding association to the various binding sites and the degree of cooperativity. Excellent reviews are available in the literature for the full and accurate analysis of protein saturation for the case of multiple binding sites (e.g., [67]). The cases of three independent binding sites (equal or different) will be briefly analysed below, calling attention to common problems when oversimplifying the analysis of the data.

There are three possibly distinct binding sites in the system (in this case all three in the same protein, but the same formalism applies if each is in a distinct protein because they are independent), so binding is described by the following scheme:(8)$$\begin{array}{l} L_{W}+B^{I}\overset{K_{b}^{I}}{↔}L_{B^{I}}\\ L_{W}+B^{\mathrm{II}}\overset{K_{b}^{\mathrm{II}}}{↔}L_{B^{\mathrm{II}}}\\ L_{W}+B^{\mathrm{III}}\overset{K_{b}^{\mathrm{III}}}{↔}L_{B^{\mathrm{III}}} \end{array}$$
where $B^{i}$ and $L_{B^{i}}$ correspond respectively to the free and occupied binding sites *i*, and $K_{b}^{i}$ is the respective microscopic association constant, with *i* representing the distinct binding sites (I, II, or III). The association of the ligand with each binding site cannot be characterized independently. Instead, a general equation considering ligand association with all binding sites must be defined and solved. If the binding sites are equivalent and independent, the solution is simply to consider the total concentration of binding sites ($\left[B\right]=\left[B^{I}\right]+\left[B^{\mathrm{II}}\right]+\left[B^{\mathrm{III}}\right]$) in Equation (2) and use the solution of the quadratic equation as indicated above. However, if the binding sites are not equivalent, the system is described by a set of *n* + 2 equations where *n* is the number of binding sites, and must be solved numerically. If the concentration of protein is known and each protein contains the three distinct binding sites, protein saturation with ligand is best described by the macroscopic overall association constants $\beta_{i}$, or the stepwise macroscopic association constants $K_{i}^{}$, defined in Equations (9) and (10), respectively.
(9)$$\begin{array}{l} L_{W}+P\overset{\beta_{1}}{↔}PL_{1};\;\beta_{1}=\frac{\left[PL_{1}\right]}{\left[P\right]\left[L_{W}\right]}\\ 2L_{W}+P\overset{\beta_{2}}{↔}PL_{2};\;\beta_{2}=\frac{\left[PL_{2}\right]}{\left[P\right]{\left[L_{W}\right]}^{2}}\\ 3L_{W}+P\overset{\beta_{3}}{↔}PL_{3};\;\beta_{3}=\frac{\left[PL_{3}\right]}{\left[P\right]{\left[L_{W}\right]}^{3}} \end{array}$$
(10)$$\begin{array}{l} L_{W}+P\overset{K_{1}^{}}{↔}PL_{1};\;K_{1}^{}=\frac{\left[PL_{1}\right]}{\left[P\right]\left[L_{W}\right]}\\ L_{W}+PL_{1}\overset{K_{2}^{}}{↔}PL_{2};\;K_{2}^{}=\frac{\left[PL_{2}\right]}{\left[PL_{1}\right]\left[L_{W}\right]}\\ L_{W}+PL_{2}\overset{K_{3}^{}}{↔}PL_{3};\;K_{3}^{}=\frac{\left[PL_{3}\right]}{\left[PL_{2}\right]\left[L_{W}\right]} \end{array}$$

Note that $PL_{i}$ represents a sub-ensemble of protein–ligand complexes with i ligands bound per protein. It does not contain information regarding the binding sites where the ligands are bound. For example, $PL_{1}$ represents the collection of protein molecules with only one ligand bound, which may be in site 1, 2 or 3. The relative fraction of ligand bound to each of the possible binding sites depends on the microscopic binding affinities ($K_{b}^{i}$). The same holds for $PL_{2}$ and $PL_{3}$, although in this case the relative fractions are not simply a result of the relative values of $K_{b}^{i}$ but also of possible dependencies between the distinct binding sites.

The two sets of macroscopic association constants (overall and stepwise) are interrelated by:(11)$$\beta_{1}=K_{1}^{}\;;\;\beta_{2}=K_{1}^{}K_{2}^{}\;;\;\beta_{3}=K_{1}^{}K_{2}^{}K_{3}^{}$$

The set of mass balance equations that needs to be solved to calculate the concentration of free ligand in solution is:(12)$$\begin{array}{l} \left[P_{T}\right]=\left[P\right]+\left[PL_{1}\right]+\left[PL_{2}\right]+\left[PL_{3}\right]=\left[P\right]+\sum_{i=1}^{n}\beta_{i}\left[P\right]{\left[L_{W}\right]}^{i}\\ \left[L_{T}\right]=\left[L_{W}\right]+\left[L_{B}\right]=\left[L_{W}\right]+\left[L_{B^{I}}\right]+\left[L_{B^{\mathrm{II}}}\right]+\left[L_{B^{\mathrm{III}}}\right]=\\ =\left[L_{W}\right]+\left[PL_{1}\right]+2\left[PL_{2}\right]+3\left[PL_{3}\right]=\left[L_{W}\right]+\sum_{i=1}^{n}i\beta_{i}\left[P\right]{\left[L_{W}\right]}^{i} \end{array}$$
where $L_{B^{x}}$ represents ligand bound to site x.

Protein saturation as a function of the concentration of free ligand is given by:(13)$$\bar{n_{P}}=\langle i\rangle =\frac{\left[L_{B}\right]}{\left[P_{T}\right]}=\frac{\sum_{i=1}^{n}i\beta_{i}\left[P\right]{\left[L_{W}\right]}^{i}}{\left[P_{T}\right]}$$
where 〈i〉 is the average number of ligands bound per protein molecule.

The ligand saturation is given by:(14)$$\bar{n_{L}}=\frac{\left[L_{B}\right]}{\left[L_{T}\right]}=\frac{\sum_{i=1}^{n}i\beta_{i}\left[P\right]{\left[L_{W}\right]}^{i}}{\left[L_{T}\right]}$$
which corresponds to the fraction of ligand molecules that is bound.

If the binding sites are independent, the stepwise macroscopic and the microscopic association constants are related by:(15)$$\begin{array}{l} K_{1}^{}=K_{b}^{I}+K_{b}^{\mathrm{II}}+K_{b}^{\mathrm{III}}\\ K_{2}^{}=\frac{K_{b}^{I}K_{b}^{\mathrm{II}}+K_{b}^{I}K_{b}^{\mathrm{III}}+K_{b}^{\mathrm{II}}K_{b}^{\mathrm{III}}}{K_{b}^{I}+K_{b}^{\mathrm{II}}+K_{b}^{\mathrm{III}}}\\ K_{3}^{}=\frac{K_{b}^{I}K_{b}^{\mathrm{II}}K_{b}^{\mathrm{III}}}{K_{b}^{I}K_{b}^{\mathrm{II}}+K_{b}^{I}K_{b}^{\mathrm{III}}+K_{b}^{\mathrm{II}}K_{b}^{\mathrm{III}}} \end{array}$$

The derivation of these equations, the relations between the stepwise macroscopic association constants for the case of dependent binding sites, and details regarding the numerical methods used to calculate the free ligand concentration may be found in the literature [4,46,62,67,68].

#### 2.2.2. Characterization of the Binding Parameters from Protein Saturation Curves

The saturation of a protein with 3 independent binding sites is shown in Figure 4 with a high affinity for binding site I ($K_{b}^{I}=10^{6}\;M^{-1}$), and distinct relations to the other binding sites to which the ligand binds with equal or lower affinity. The total concentration considered for the protein is 10 µM and the total ligand concentration is varied to cover the full range of protein saturation.

When the affinities for the distinct binding sites differ by 3 orders of magnitude (upper plots), saturation of the distinct sites occurs at very different ligand concentrations (plot A), and proteins associate with 1, 2 or 3 ligands at different ligand concentrations (plot B), leading to a saturation curve with well-defined steps (plot C). A similar behaviour is observed for binding affinities that differ by 2 orders of magnitude (plots D to F), although the stepwise saturation of protein with ligand becomes less pronounced, with extensive overlap of the ligand concentration range for formation of $PL_{1}$, $PL_{2}$, and $PL_{3}$. This overlap is even more pronounced when the affinities differ by only one order of magnitude (plots G to I), in which case the saturation steps can no longer be observed in the protein saturation curve. The lower plots show the case where all 3 binding sites in the protein have equal affinity for the ligand. In this case, proteins with only 1 or 2 ligands have a relatively low abundance, and over a very limited ligand concentration range. In this case, the narrowness of the ligand concentration range where the proteins have intermediate numbers of ligands bound arises from the following two reasons. First, taking each binding site from 10 to 90% saturation involves nearly two orders of magnitude variation in $\left[L_{W}\right]$, from 0.11 $K_{d}$ to 9 $K_{d}$. When the binding affinities are very different, significant occupation of each binding site occurs at very distinct ligand concentrations, leading to a wide concentration range for protein saturation. In turn, for similar binding affinities, the concentration ranges of significant occupation with ligand overlap. Second, for similar binding affinities and intermediate values of the average protein saturation, it is equally probable that the ligand is associated with different binding sites in the same protein, or to the same or a different type of binding site in distinct proteins, thus decreasing the relative population of protein molecules with intermediate values of ligands bound.

The best fit of the saturation curves with a model considering an adjustable number of equal and independent binding sites per protein (solution $x_{-}$ of Equation (3)) is shown in the plots on the right. As expected, a perfect fit and the correct binding parameters (*n* = 3 and *K*_b_ = 10^6^ M^−1^) are obtained when the binding affinity for the distinct binding sites is the same (plot L). On the other extreme, if the distinct binding sites have well separated ligand affinities, this simplified model leads to clear misfits to the saturation curve, with no risk of accidental inadequate use (plots C and F). However, the full description of the protein saturation curve requires varying the ligand concentration over 8 orders of magnitude. Such a wide range of ligand concentrations is not usually evaluated experimentally. If only 2 or 3 orders of magnitude are considered, only the binding of the first ligand would be observed, leading to a good fit. For binding affinities differing by only an order of magnitude (plot I), the stepwise association is not evident in the shape of the saturation curve, which becomes reasonably well described by the simplified model considering equal and independent binding sites. The best fit leads to a similar value for the number of binding sites, but to an intermediate value of the association constant. The simulations shown in Figure 4 assume that the protein concentration is well known and that protein saturation may be quantitatively and directly followed. Obviously, the model that assumes one binding site per protein is not adequate to describe the saturation curve in this case (dashed grey lines).

The analysis of the simulations depicted in Figure 4 shows that when the number of ligands per protein may be directly obtained, the analysis of the results with a model considering a variable number of equal and independent binding sites is a good first option. If the model adequately describes the shape of the saturation curve (plots I and L in this case study), the association constant estimated corresponds to the microscopic association constant if the binding sites are equal and independent. If the binding affinity for the binding sites is different, the value obtained is close to the average value (within the same order of magnitude) of all association constants. If the model does not adequately describe the shape of the saturation curve, the values obtained are therefore devoid of physical meaning, and more complicated models must be used [56,62,63,64,65,67,68,69,70]. The accurate description of the saturation curve permits obtainment of the global association constants, Equation (9), and the stepwise macroscopic association constants, Equation (10). If the binding sites are independent, the number of microscopic and macroscopic binding constants is the same and it may in principle be possible to obtain one set of constants from the other. However, if the binding sites are not independent, the number of microscopic binding constants exceeds the number of parameters that may be obtained from the protein saturation curves, and therefore cannot be quantitatively characterized. In this case, only some relations between them may be obtained, as well as possible indication of cooperativity for binding to the distinct binding sites.

#### 2.2.3. Scatchard Plots and Evaluation of Cooperativity

The analysis of the protein saturation curve with complex models requires the use of complicated fitting procedures, including numerical methods to obtain the concentration of free ligand and protein, the equivalent to Equation (12) but adjusted to the number of binding sites. To avoid the laborious procedure of data analysis (or, in an attempt to complement the analysis), specific graphical representations of the data are usually performed to evaluate for the presence of cooperativity between the binding sites. Scatchard plots are a prominent example, where $\bar{n_{P}}/\left[L_{W}\right]$ is represented as a function of $\bar{n_{P}}$. A linear dependence between both variables is expected for the case of a protein with n independent binding sites, and deviations from linearity are interpreted as indicative of cooperativity.
(16)$$\frac{\bar{n_{P}}}{\left[L_{W}\right]}=nK_{b}-\bar{n_{P}}K_{b}$$

Before analysing the results obtained with Scatchard plots, some remarks must be made regarding this and other linearization procedures. A first big limitation is that, because the binding constants are still not known (the objective is to obtain them from this analysis), if the concentrations of bound or free ligand are not directly measured, $\left[L_{W}\right]$ cannot be calculated. In these situations, $\left[L_{T}\right]$ is commonly used, leading to incorrect interpretations. Another important concern is that these linearization procedures do not comply with the assumptions imposed by non-linear least-square regression analysis, e.g., that the x-axis must be an independent variable with negligible error. Also, while the uncertainty associated with concentrations may be assumed to be normally distributed and to be constant within the range analysed (allowing the use of regressions not weighted by the uncertainty in the variables represented in the x and y axis), this is not the case for the inverse of concentration (which shows a much higher uncertainty for large values of the variable, corresponding to low concentrations). The confidence intervals are also not symmetric for inverse concentrations, with both properties leading to best fits that do not reflect the average behaviour of the system [71]. Finally, those linearization procedures were introduced at a time when computers were not available for data analysis. While the risk of misinterpreting the data through the use of linearization procedures was acceptable then, it is certainly not justified now.

To further evaluate the adequacy of data linearization and representation in Scatchard plots, this procedure was applied to the case of 3 independent binding sites in the conditions considered previously for Figure 4. The results are shown in Figure 5. Plot A represents the protein saturation divided by the concentration of free ligand, as a function of protein saturation. As expected, when the ligand affinity to all binding sites is the same (black), a linear dependence is observed with the intercept in the y-axis being equal to $n\;K_{b}$ (3 × 10^6^ M^−1^), the x-axis intercept being n, and the slope equal to $K_{b}$. In the other extreme, when the binding affinities differ by 3 orders of magnitude (blue), only the population of the highest affinity site is captured (although in the concentration range analysed they all become occupied, as indicated in the x-axis that goes up to $\bar{n_{P}}$ = 3), with a linear variation leading to $n\;K_{b}$(1.0 × 10^6^ M^−1^) and n = 1.0. A similar behaviour is observed when the affinities differ by two orders of magnitude (green), the best linear fit for low values of $\bar{n_{P}}$ leading to $n\;K_{b}$(9.9 × 10^5^ M^−1^) and n = 1.0. When the binding affinities differ by only one order of magnitude (red), non-linear Scatchard plots are obtained, with the best fit for small $\bar{n_{P}}$ leading to $n\;K_{b}$(9.0 × 10^5^ M^−1^) and n = 1.2. The non-linear dependence may be incorrectly interpreted as negative cooperativity, despite resulting from binding to different, but independent, binding sites. Therefore, unless the protein being studied is known to have several equivalent binding sites (as is usually the case for multimeric proteins) and the sample is of high purity, negative cooperativity cannot be inferred from non-linear Scatchard plots.

As expected, if the total concentration of ligand is used instead of the free concentration (plot B), the information obtained is meaningless and misleading. In this case, non-linear dependences are obtained for all cases analysed, with the obtained downward curvatures suggesting positive cooperativity. Unfortunately, this incorrect data analysis may be found in the literature. Hopefully this review will alert inexperienced readers to possible pitfalls and will contribute to a proper data analysis.

#### 2.2.4. Indirect Characterization of the Binding Parameters Indirectly from Changes in a Protein Property

In the previous analysis, it was assumed that the number of ligands per protein could be directly characterized. However, if protein saturation is not directly accessed, but rather inferred from a property that varies with protein saturation (ΔS), the number of binding sites can only be evaluated from the analysis of the results, and inadequate procedures may lead to an incorrect description of the system. This will be addressed in the current section, considering proteins with 3 independent binding sites and several cases with distinct relative values of the binding affinities.

The signal from the protein at a given concentration of ligand, S, depends on the concentration of all different proteins species, $\left[PL^{i}\right]$, and on the signal characteristic of each species, $s_{i}$, Equation (17).
(17)$$\begin{array}{ll} \Delta S & =S-S_{0}=s_{0}\left[P\right]+\sum_{i=1}^{N}s_{i}\left[PL_{}^{i}\right]-s_{i}\left[P_{T}\right]\\ & =\sum_{i=1}^{N}\left(s_{i}-s_{0}\right)\left[PL_{}^{i}\right] \end{array}$$

Note that $PL_{}^{i}$ does not represent a protein with *i* ligands, which would be $PL_{i}^{}$, in the notation followed in this work. A protein with a given number of ligands corresponds to many distinct species, depending on the binding sites to which the ligands are bound. Thus, in Equation (17), *N* is not equal to the number of binding sites (n), but rather to the number of possible combinations of bound ligands. Assuming that the same signal is obtained for all combinations with the same number of bound ligands, Equation (17) becomes
(18)$$\Delta S=\sum_{i=1}^{n}\left(s_{i}-s_{0}\right)\left[PL_{i}^{}\right]$$
and may be expressed in terms of the macroscopic binding associations, Equation (19).
(19)$$\Delta S=\sum_{i=1}^{n}\left(s_{i}-s_{0}\right)\beta_{i}\left[P\right]{\left[L_{W}\right]}^{i}$$

In this section, it is considered that the signal variation is the same when the ligand binds to any of the binding sites, and that the magnitude of the signal variation is proportional to the number of bound ligands, $\left(s_{i}-s_{0}\right)=\Delta s_{\max}\left(i/n\right)$. This is a good approximation when properties such as the overall mass or charge of the ligand–protein complex are being followed. However, the observed property may vary differently, and this aspect will be discussed in Section 2.3 in the context of quenching of protein fluorescence by bound ligand.

From the analysis of the signal variation as a function of the ligand concentration, it is in principle possible to obtain the intrinsic signal variation for each number of bound ligands $\left(s_{i}-s_{0}\right)$, and the macroscopic association constants $\beta_{i}$. In turn, from the relations between the macroscopic association constants, some information may be obtained regarding the similarity and/or dependency of the different binding sites, Equation (20).
(20)(βi(in))1i(βi−1(i−1n))1i−1=1 identical and independent binding sites<1 non-identical binding sites    or identical but not independent (negative cooperativity) >1 non-identical binding sites    or identical but not independent (positive cooperativity) 

In the case of equal and independent binding sites, the macroscopic association constants depend on the microscopic association constant according to [4]:(21)$$\begin{array}{l} \beta_{i}=\prod_{i=1}^{n}K_{i}\\ K_{i}=\frac{n-i+1}{i}K_{b} \end{array}$$

The simulations shown in Figure 6A were performed for $\left[P_{T}\right]$ = 1 µM, $K_{b}^{I}$ = 10^6^ M^−1^, and at different relative values of the binding affinity to the other two binding sites in the protein ($K_{b}^{I}/K_{b}^{\mathrm{II}}=K_{b}^{\mathrm{II}}/K_{b}^{\mathrm{III}}=10$ in blue; $K_{b}^{I}/K_{b}^{\mathrm{II}}=K_{b}^{\mathrm{II}}/K_{b}^{\mathrm{III}}=3$ in green, and $K_{b}^{I}/K_{b}^{\mathrm{II}}=K_{b}^{\mathrm{II}}/K_{b}^{\mathrm{III}}=1$ in red). The lines correspond to the best fit of Equation (3) assuming an adjustable number of equal and independent binding sites (*n* ≥ 1, solid thin lines), or considering only one binding site per protein (*n* = 1, dashed thick lines). The two sets of fitting curves cannot be visually distinguished and lead to the same parameters for the cases considered of binding sites with different affinities (curves in blue and green). The quality of the best fit for different values of *n* is presented in Appendix A, showing a small improvement as *n* decreases. The fitting parameters are thus the same for both models, with the association constant obtained from the best fit being similar to that of the binding site with intermediate affinity ($K_{b}^{\mathrm{II}}$ = 10^5^ M^−1^) and *n* = 1. However, different parameters are obtained for the best fit of the simulation results when all binding sites are equal (Figure 6, curve and parameters in red). In this case, the best fit with adjustable *n* shows a minimum in the squared deviation for *n* = 3 (plot B), with reasonable sensitivity for both lower and higher values of *n*. Nevertheless, the representation of the best fit curves for values of *n* from 1 to 4 (plot C) shows that the fitting sensitivity is insufficient for the accurate characterization in the case of experimental data due to the unavoidable presence of noise (here represented as large symbols for the simulated results).

In all cases considered, the magnitude of the signal variation per ligand bound decreases as the number of binding sites considered increases, such that the total variation in the signal is obtained only when the protein is fully saturated (Figure 6B and Appendix A, plots A and C). Somewhat surprisingly, the association constant obtained from the best fit decreases with the decrease in the number of binding sites, leading to a large variation in the overall protein binding affinity obtained from the best fit: $nK_{b}$ varies from 5.4 × 10^5^ M^−1^ for *n* = 1 to 5.9 × 10^6^ M^−1^ for *n* = 4 in the case of equal binding sites, with the correct description of the system for *n* = 3 leading to $nK_{b}$ = 3.0 × 10^6^ M^−1^. However, the quality of the distinct best fits to experimental results would not allow a discrimination between the distinct set of parameters, and the simpler model would thus be selected. This would underestimate the global binding affinity by nearly one order of magnitude. Attempting to understand this effect, distinct simulations have been performed at lower and higher affinities. The results are shown in Appendix A. For $K_{b}^{I}$ = 10^4^ M^−1^ (plots A to C), the quality of the best fit with a variable number of equal and independent binding sites is even less sensitive to *n*, with $K_{b}$ being always approximately 1 × 10^4^ M^−1^ for the case of equal binding sites (plot B and C), and close to the average $K_{b}^{i}$ for different affinity binding sites. In the case of high-affinity binding, $K_{b}^{I}$ = 10^8^ M^−1^ (plot D), a high sensitivity to *n* is observed for the case of equal binding sites (plots E and F) and for $K_{b}^{I}/K_{b}^{\mathrm{II}}=K_{b}^{\mathrm{II}}/K_{b}^{\mathrm{III}}=3$ (plots G and H), but not for $K_{b}^{I}/K_{b}^{\mathrm{II}}=K_{b}^{\mathrm{II}}/K_{b}^{\mathrm{III}}=10$ (plots I and J).

The analysis of the data presented in Figure 6 and Appendix A shows that the sensitivity of the best fit to n depends on the fraction of ligand associated with the protein, which is presented in Figure 7 for the various conditions considered. If the ligand is in large excess, the shape of the protein saturation curve is not dependent on the considered number of binding sites. In this case, it is not possible to obtain the number of binding sites from the best fit of the saturation curve. On the other hand, if the fraction of ligand bound to the protein is significant, increasing the number of binding sites leads to a decrease in the fraction of ligand free in solution, and consequently to differences in the shape of the protein saturation curve. In this case, the protein saturation curve is a straight line until almost full saturation. When the data is tentatively described by a model with fewer binding sites, the predicted fraction of free ligand is higher and the saturation curve linearly reaches saturation at lower total ligand concentrations. The best fit then obtained underestimates saturation at high values and overestimates it at low saturation, leading to a lower association constant.

In conclusion, when protein saturation with ligand is followed indirectly for proteins with several similar binding sites, information regarding the number of binding sites and their respective association constants can only be obtained if binding is limited by ligand availability. Otherwise, the dependence of the property on the total ligand concentration follows a hyperbola and is insensitive to the number of binding sites in the protein. In this situation, there is no justification to use complex models, and the simplest one (*n* = 1) is usually used. The association constant obtained is similar to or smaller than the true binding association, thus leading to a significant underestimation of the fraction of ligand bound to the protein. If binding is limited by ligand availability but the data is noisy, it will also be impossible to distinguish between models with different *n*. The description of the system with the simplest model may in this case lead to a severe underestimation of the amount of ligand associated with the protein. It is therefore essential to perform the experiment at different protein concentrations, to check for consistency and find the most adequate model and parameter values.

### 2.3. Quenching of Protein Fluorescence: Stern–Volmer Plots

The characterization of binding to proteins through ligand quenching of protein fluorescence is a method that is very commonly used and deserves special attention. The fluorescence decrease is usually described by the Stern–Volmer equation,
(22)$$\frac{S_{0}}{S}=1+K_{\mathrm{SV}}\left[L_{W}\right]$$
where $S_{0}$ and S represent the fluorescence intensity in the absence and in the presence of ligand, respectively. The latter depends on the concentration of all different protein species, $\left[PL_{}^{i}\right]$, and on the signal characteristics of each species, $s_{i}$, Equation (17) in Section 2.2.4.

The equation for $S_{0}/S$ is thus:(23)$$\frac{S_{0}}{S}=\frac{s_{0}\left[P_{T}\right]}{s_{0}\left[P\right]+\sum_{i=1}^{N}s_{i}\left[PL_{}^{i}\right]}$$

Remember that $PL_{}^{i}$ does not represent a protein with *i* ligands, but rather each different combination of i ligands bound to the distinct binding sites. Thus, in Equations (17) and (23) *N* is not equal to the number of binding sites (n), but rather to the number of possible combinations of bound ligands. Assuming that the same quenching efficiency is obtained for all combinations with the same number of bound ligands, Equation (23) may be expressed in terms of the macroscopic binding associations, where N is now equal to n:(24)$$\frac{S_{0}}{S}=\frac{\left[P_{T}\right]}{\left[P\right]+\sum_{i=1}^{n}\frac{s_{i}}{s_{0}}\left[PL_{{}_{i}}^{}\right]}=\frac{\left[P\right]+\sum_{i=1}^{n}\beta_{i}\left[P\right]{\left[L_{W}\right]}^{i}}{\left[P\right]+\sum_{i=1}^{n}\frac{s_{i}}{s_{0}}\beta_{i}\left[P\right]{\left[L_{w}\right]}^{i}}=\frac{1+\sum_{i=1}^{n}\beta_{i}{\left[L_{W}\right]}^{i}}{1+\sum_{i=1}^{n}\frac{s_{i}}{s_{0}}\beta_{i}{\left[L_{w}\right]}^{i}}$$

It should be highlighted that the validity of this approximation is difficult to evaluate and will depend both on the protein and ligand. For example, for proteins with several fluorescent groups (e.g., bovine serum albumin, which has two tryptophan residues), the fluorescence quantum yield of each group may be different, and each of them will most likely be quenched with different efficiencies by ligands in the distinct binding sites. This approximation is also invalid if ligand binding at some of the binding sites in the protein causes conformational changes that affect the protein fluorescence quantum yield and quenching efficiency by subsequent ligands. In the following analysis the possible non-validity of this approximation will be ignored, but this should be kept in mind whenever the results obtained by protein fluorescence quenching do not agree with those from other approaches.

Equation (24) is further simplified if it may be considered that the residual protein fluorescence is zero when any of the binding sites is occupied with ligand ($s_{i}=0\;for\;i\neq 0,$ corresponding to 100% efficient fluorescence quenching by the ligand):(25)$$\frac{S_{0}}{S}=\frac{\left[P_{T}\right]}{\left[P\right]}=\frac{\left[P\right]+\sum_{i=1}^{n}\beta_{i}\left[P\right]{\left[L_{W}\right]}^{i}}{\left[P\right]}=1+\sum_{i=1}^{n}\beta_{i}{\left[L_{W}\right]}^{i}$$

Although usually assumed, full quenching of the protein fluorescence by the ligand is frequently not supported experimentally, and in fact it is commonly not expected. Attention will therefore be given to the consequences of its non-validity later in this section.

The relation between the Stern–Volmer constant and the binding constants to the different binding sites may be obtained by comparing Equation (25) with Equation (22), leading to:(26)$$K_{\mathrm{SV}}=\frac{\sum_{i=1}^{n}\beta_{i}{\left[L_{w}\right]}^{i}}{\left[L_{w}\right]}$$

If the protein has only one binding site, Equation (26) simplifies to Equation (27), and the Stern–Volmer constant corresponds to the association constant:(27)$$K_{\mathrm{SV}}=\beta_{1}=K_{b}$$

Thus, for proteins with a single binding site, when ligand binding leads to full quenching of protein fluorescence, and if the free (not the total) ligand concentration is used as the independent variable, the slope of a linear Stern–Volmer plot is equal to the association binding constant. If any of the above conditions is violated, $K_{\mathrm{SV}}$ will deviate from $K_{b}$.

If the proteins contain several binding sites, there is no simple correspondence between $K_{\mathrm{SV}}$ and $K_{b}$. In fact, in this case $K_{\mathrm{SV}}$ is not a constant, its value is dependent on the concentration of ligand, and thus non-linear Stern–Volmer plots are obtained. This is frequently observed when characterizing ligand association with proteins containing several binding sites, such as serum albumin. Although this is the expected behaviour, the authors usually analyse the results with the simplified Equation (22) which is unable to capture the observed behaviour and leads to misinterpretation. However, some relevant information may be obtained when the linear range of $S_{0}/S$ variation is considered (that is, for low ligand concentrations). In this case, Equation (26) simplifies to Equation (28). Therefore, in the case of proteins with several binding sites with full quenching by the ligand in any of the binding sites, the value of $K_{\mathrm{SV}}$ obtained from the linear fit at low ligand concentrations provides an estimate of the overall binding affinity.
(28)$$K_{\mathrm{SV}}\left(\mathrm{small}\;\left[L_{W}\right]\right)=\beta_{1}=\sum_{i=1}^{n}K_{b}^{i}$$

In the simplest case, with n identical and independent binding sites, $\beta_{1}=n\;K_{b}$.

If the quenching efficiency is not 100%, the limit at low ligand concentrations may still provide useful information. From the general Equation (23) one may obtain Equation (29). The derivation of Equations (28) and (29) is provided in Appendix A.
(29)$$K_{\mathrm{SV}}\left(small\;\left[L_{W}\right]\right)=\sum_{i=1}^{n}\;\frac{s_{0}-s_{i}}{s_{0}}\;K_{b}^{i}$$

The quenching efficiency is usually an unknown parameter, which prevents researchers from using Equation (29). If Equation (28) is used but the quenching efficiency is not 100%, the value obtained for the parameter $\beta_{1}$ will underestimate the true value. For example, if the quenching efficiency is only 10% ($s_{i}=0.9$), the estimated value for $\beta_{1}$ will be an order of magnitude lower than its true value. Higher quenching efficiencies lead to a more accurate estimation of $\beta_{1}$.

To exemplify some expected Stern–Volmer plots for quenching of a protein with several binding sites, a set of conditions has been simulated for ligand binding to a protein with 3 binding sites, Figure 8 and Figure 9. In Figure 8 the 3 binding sites are considered equal and independent, with the ligand affinity changing from low ($K_{b}^{}=10^{4}\;M^{-1}$), to intermediate ($K_{b}^{}=10^{6}\;M^{-1}$), and high ($K_{b}^{}=10^{8}\;M^{-1}$). However, the possibility of different quenching efficiencies for the ligand bound to each binding site was accounted for in the examined situations: 100% quenching efficiency regardless of the binding site to which the ligand is associated ($s_{i}=0$ for *i* = *I* to *III*, circles in black), $s_{I}$ = 0.6, $s_{\mathrm{II}}$ = 0.3, and $s_{\mathrm{III}}$= 0 (circles in red), and only ligand in binding site *III* leading to fluorescence quenching $s_{I}$ = $s_{\mathrm{II}}$ = 1, and $s_{\mathrm{III}}$ = 0 (circles in blue). For the case of distinct quenching efficiencies, it was considered that when two binding sites are occupied the fluorescence quenching efficiency is given by the product of the residual fluorescence when the ligand is in each of the binding sites ($s_{I\&\mathrm{II}}=s_{I}s_{\mathrm{II}}$). The concentration of protein with a given combination of ligands $\left(\left[PL_{}^{i}\right]\right)$ was calculated from $\left[PL_{i}\right]$ and the relative value of the binding affinities that lead to that ligand occupation. As an example, the concentration of protein with ligand only in binding site *I* is equal to $\left[PL_{1}\right]K_{b}^{I}/\left(K_{b}^{I}+K_{b}^{\mathrm{II}}+K_{b}^{\mathrm{III}}\right)$, and with ligand in binding sites *I* and *II* is equal to $\left[PL_{2}\right]\left(K_{b}^{I}+K_{b}^{\mathrm{II}}\right)/\left(2\left(K_{b}^{I}+K_{b}^{\mathrm{II}}+K_{b}^{\mathrm{III}}\right)\right)$.

The first observation from the results shown in Figure 8 is that Stern–Volmer plots with an upward curvature are generally obtained. The sole exceptions occur when only the ligand bound to site III leads to fluorescence quenching (blue circles). This situation is formally equivalent to having only one binding site (i.e., the other binding sites are “silent”), and in this case a linear dependence of $S_{0}/S$ with $\left[L_{W}\right]$ is in fact expected, Equation (27). However, if (incorrectly) the total concentration of ligand is considered (bottom plots), an upward curvature is observed even in this case when the binding affinity is intermediate or high (plots E and F).

Due to the non-linear Stern–Volmer plots, the value of $K_{\mathrm{SV}}$ obtained from the best fit of the simplified Equation (22) depends on the considered range of ligand concentrations and the corresponding magnitude of fluorescence variation. The dashed lines shown in Figure 8 were obtained from the best fit for $S_{0}/S$ between 1 and 1.1. When the concentration of free ligand is considered (upper plots), the values obtained for $K_{\mathrm{SV}}$ lead to reasonable estimates of $K_{b}$ (within the same order of magnitude) for all considered conditions, in agreement with the predictions from Equations (28) and (29). As predicted from Equation (29), the value of $K_{\mathrm{SV}}$ depends on the extent of protein fluorescence quenching by the ligand. The larger the quenching efficiency, the higher the value obtained for $K_{\mathrm{SV}}$. The upper limit of $K_{\mathrm{SV}}$ is obtained for $s_{i}=0,\;i\neq 0$ and should have been equal to $K_{b}^{I}+K_{b}^{\mathrm{II}}+K_{b}^{\mathrm{III}}$, Equation (28). However, somewhat higher values are obtained in some situations due to the upward curvature of $S_{0}/S$.

The above discussion shows that the order of magnitude of the binding affinity may be obtained from the analysis of the linear region of Stern–Volmer plots (Equations (22), (27), (28) or (29), depending on the specific conditions) if the concentration of free ligand is known. However, unless measured directly, the concentration of free ligand cannot be known before calculation of the binding affinity. If the total concentration of ligand is considered instead, deviations by several orders of magnitude may be obtained (e.g., for $K_{b}=10^{8}M^{-1}$; $K_{\mathrm{SV}}\left(\left[L_{T}\right]\right)$ is 1.2 × 10^6^ M^−1^ if $s_{i}=0,\;i\neq 0$, 8.1 × 10^5^ M^−1^ if $s_{I}=0.6,\;s_{\mathrm{II}}=0.3,\;s_{\mathrm{III}}=0$, and 4.1 × 10^5^ M^−1^ if $s_{I}=s_{\mathrm{II}}=1,\;s_{\mathrm{III}}=0$).

To increase the accuracy of $K_{\mathrm{SV}}$ as an estimate of the protein-binding ability, its evaluation must include the calculation of the corresponding free ligand concentrations. This approach is not feasible for proteins with several binding sites with distinct ligand affinities and/or binding cooperativity, because in these cases $\left[L_{W}\right]$ must be calculated numerically for each evaluation of $K_{\mathrm{SV}}$ (and corresponding $K_{b}$). A reasonable compromise between accuracy and feasibility may be achieved by considering that all binding sites are equal and independent. In this case, $\left[L_{W}\right]$ may be calculated from $\left[L_{T}\right]-\left[L_{B}\right]$, with $\left[L_{B}\right]$ being the $x_{-}$ solution of Equation (3). In this case (considering equal and independent binding sites), the binding affinity for the binding sites ($K_{b}$) may be obtained from the dependence of $S_{1}/S$ vs. $\left[L_{W}\right]$ through several alternative approaches: (i) $K_{\mathrm{SV}}$ may be obtained from the best fit of a straight line at low quenching, $S_{1}/S$ between 1 and 1.1 being usually a good balance between sensitivity and little deviation from linearity. The binding constant $K_{b}$ is then obtained from $K_{\mathrm{SV}}$ using Equation (28), considering 100% quenching efficiency ($s_{i}=0,\;i\neq 0$), or (29) if the quenching efficiency is known. In this approach it is assumed that each protein has n equal and independent binding sites, with the corresponding $\left[B_{T}\right]$ in Equation (3) being given by $n\left[P_{T}\right]$. (ii) $K_{b}$ may also be obtained from the best fit of the whole range of $S_{1}/S$, assuming that the same quenching efficiency is attained for proteins with 1 or more ligands bound ($s_{i}/s_{0}=s_{x}/s_{0},\;i\neq 0$), and with $S_{1}/S$ given by Equation (30).
(30)$$\frac{S_{0}}{S}=\frac{1+\sum_{i=1}^{n}\beta_{i}{\left[L_{W}\right]}^{i}}{1+\frac{s_{x}}{s_{0}}\sum_{i=1}^{n}\beta_{i}{\left[L_{W}\right]}^{i}}$$
with $\beta_{i}$ being related with $K_{b}$ through Equation (21) in Section 2.2.4, for the case of n equal and independent binding sites.

A Microsoft Excel file has been developed for the analysis of Stern–Volmer-like plots for protein–ligand binding experiments and will be provided to interested readers upon request.

The fitting procedure indicated above was applied to the simulated data to evaluate its applicability limits. For low binding affinity ($K_{b}$ = 10^4^ M^−1^) and 100% quenching efficiency, the best linear fits to $S_{1}/S$ < 1.1 lead to good estimates of $n\;K_{b}$, independently of the number of binding sites considered (n = 1 to 4). When the whole curve was analysed, assuming n = 1 leads to a poor fit and overestimation of $n\;K_{b}$. A better fit and more accurate parameter estimates were obtained for n = 3, suggesting that this analysis may provide some information on the number of binding sites *per* protein. As observed before (Figure 8), as the quenching efficiency is progressively decreased, $n\;K_{b}$ tends to be underestimated, both from the linear best fit at low $S_{1}/S$ or from the whole range of $S_{1}/S$. An equivalent situation is obtained for intermediate binding affinities ($K_{b}$ = 10^6^ M^−1^), with the linear best fit leading to good estimates and some discrimination regarding the number of binding sites when the whole curve is analysed. For high binding affinities and 100% quenching efficiency, only n = 1 could be used for the best linear fit to $S_{1}/S$ < 1.1. Setting n > 1 leads to poor sensitivity and estimates of $n\;K_{b}$ diverging to infinity. This is due to the upward curvature that occurs even at very low quenching. Again, as the quenching efficiency is decreased, the estimates obtained for $n\;K_{b}$ deviate significantly from the true values, underestimating this parameter by more than two orders of magnitude for $s_{I}=s_{\mathrm{II}}=1,\;s_{\mathrm{III}}=0$. The values obtained for the parameters of all conditions considered are shown in the Appendix A. Importantly, the previous results show that the analysis of Stern–Volmer plots per se underestimates the values of high-affinity binding constants, and thus cannot be relied upon for their estimation. For high-affinity binding, the direct quantification of the free ligand concentration is required. This allows understanding of the observation that the binding affinities obtained from the analysis of Stern–Volmer plots are never much higher than 10^6^ M^−1^, leading to dependences of $n\;K_{b}$ for homologous series (e.g., with increased hydrophobicity) that level-off close to 10^6^ M^−1^ [72].

In the above analysis, the models used considered that the binding sites in the protein were equal and independent. This was justified when describing the simulated results as they were in fact obtained for equal and independent binding sites. This is usually not the case in real systems where the complexity of the system may be significantly higher, with the binding sites in the protein being different and/or displaying cooperativity. If the general Equation (25) is used without any assumption regarding the relation between the macroscopic binding constants, from their relations the dissimilarity of the binding sites or eventual cooperativity in ligand binding could in principle be evaluated, Equation (20) in Section 2.2.4. However, this procedure is strongly discouraged because it could lead to severe misinterpretation of the data. To elucidate this, the simulations obtained assuming that the protein contains 3 equal and independent binding sites with $K_{b}=10^{6}\;M^{-1}$ and a fluorescence quenching efficiency dependent on the binding site $\left(s_{1}=0.6,\;s_{2}=0.3,\;s_{3}=0\right)$ was analysed with several possible combinations of $\beta_{i}$. The results obtained are shown in Figure 9.

As expected, the variation of $S_{0}/S$ predicted from Equation (25) for the set of binding parameters used to obtain the simulation, n = 3 and $K_{b}^{i}$ = 1.0 × 10^6^ M^−1^ for all i (green line on plot A), does not accurately describe the simulation results. This is because only partial fluorescence quenching was considered, $s_{1}=0.6,\;s_{2}=0.3,\;s_{3}=0$, which is not in agreement with the approximations required to obtain Equation (25). To obtain a good fit it is necessary to adjust the binding parameters which will thus deviate from the “true” ones. Many different combinations of parameter estimates yield equally good fits of the observed dependence of $S_{0}/S$ with $\left[L_{W}\right].$ The examples shown in Figure 9 correspond to the best fit of equation (25), allowing $\beta_{i}$ to vary while fixing the number of binding sites at n = 3 (plot A), n = 2 (plot B), or n = 1 (plot C). The best fit with n as an adjustable parameter and considering infinite cooperativity, a procedure commonly followed to describe Stern–Volmer plots with upward curvature [53,72], is also shown in plot C. The ratio between the values of the concentration of free ligand obtained from each best fit, ${\left[L_{W}\right]}_{\mathrm{Fit}}$, and its “true” value, is shown in plots D to F. For this particular situation this ratio is almost always higher than 1, to compensate for the apparent weaker binding affinity that results from the inefficient fluorescence quenching. The values obtained for $\beta_{i}$ vary widely, suggesting positive cooperativity in some cases and negative cooperativity in others, with the binding affinity ranging from ≅ 10^6^ M^−1^ (close to the true one) to over 10^8^ M^−1^.

This shows very clearly that detailed information regarding the binding properties of proteins with multiple binding sites cannot be obtained from Stern–Volmer plots of protein fluorescence quenching by the ligand. At most, the order of magnitude of the overall binding affinity may be obtained. It is therefore preferable to consider the simplest models, with only one binding site per protein, or several equal and independent binding sites at most.

The simulations shown in this section considered proteins with several binding sites, and it was shown that an upward curvature in the Stern–Volmer plots is expected, Figure 8 and Figure 9. The remainder of this section will be dedicated to the simpler case of proteins with a single binding site, and it will be seen that in this case downward curvature is the general behaviour. A downward curvature in Stern–Volmer plots is commonly observed in protein-binding experiments, and it is usually interpreted as indication that there are several fluorophores in the protein and not all are being quenched by the ligand. For the case of proteins containing several tryptophan residues this is a particularly convenient interpretation. However, the reason behind this behaviour may be simply that the presence of the ligand in the binding site does not lead to full quenching ($s_{i}\neq 0$).

Figure 10 shows simulations for a protein with one binding site, that binds the ligand with low, moderate or high affinity, leading to full or only partial quenching of the protein fluorescence. The simulation results for $S_{0}/S$ were analysed to obtain estimates of the binding affinity and fluorescence quenching efficiency. The equations used for the linear best fit at small $S_{0}/S$ and for the best fit of the whole variation in $S_{0}/S$ are easily obtained from Equations (29) and (30), respectively, and are shown below for completeness.

The slope at $S_{0}/S$ close to unity is related to the binding affinity and quenching efficiency by:(31)$$K_{\mathrm{SV}}\left(\mathrm{small}\;\left[L_{W}\right]\right)=\left(1-\frac{s_{1}}{s_{0}}\right)\;K_{b}^{}$$
and the whole variation of $S_{0}/S$ is related to the binding affinity and quenching efficiency by:(32)$$\frac{S_{0}}{S}=\frac{1+K_{b}\left[L_{W}\right]}{1+\frac{s_{1}}{s_{0}}K_{b}\left[L_{W}\right]}$$

Equation (32) is formally equivalent to the equation commonly used to account for downward curvature in Stern–Volmer plots, with $s_{1}/s_{0}$ being interpreted as the fraction of fluorescent groups in the protein that are not accessible to the ligand ($1-f_{a}$). However, the interpretation of the results obtained from both approaches is fundamentally different. In Equation (32), $s_{1}/s_{0}$ provides information regarding the proximity of the ligand to the protein fluorophores in a single binding site protein. In contrast, in the usual analysis, $f_{a}$ is interpreted as the fraction of ligands in a given binding site, assuming *a priori* and without any support that quenching is 100% efficient and that the protein has multiple binding sites.

In Figure 10, the quenching efficiency was varied from 100% to 20% ($s_{1}/s_{0}$ = 0, 0.2, 0.5, and 0.8, left to right plots), and the binding affinity was varied from low to high ($K_{b}$ = 10^4^, 10^6^, and 10^8^ M^−1^, top to bottom). The dashed lines in each plot correspond to the best fit of Equation (22) for $S_{0}/S$ close to unity, with the relation of $K_{\mathrm{SV}}$ with $K_{b}$ given by Equation (31), and the concentration of free ligand calculated from the $x_{-}$ solution of Equation (3). The circles and dashed lines in red assume full quenching, while the value of $s_{1}/s_{0}$ considered in the simulations is used for the circles and dashed lines in blue. To highlight the inadequacy of using $\left[L_{T}\right]$ instead of $\left[L_{W}\right]$, this condition is also shown with black circles and dashed lines. Finally, the best fit of Equation (32) to the whole variation of $S_{0}/S$ is shown as green circles and a continuous line. In this case, the quenching efficiency was an adjustable parameter. The estimated parameter obtained from the distinct best fits is shown in the respective plots with the same colours as the corresponding curves.

When the data is analysed with the best fit of the linear region, only one parameter may be estimated, $K_{\mathrm{SV}}$. If the quenching efficiency is known, $K_{b}$ may be calculated accurately from $K_{\mathrm{SV}}$ using Equation (31), otherwise $K_{b}$ will be underestimated. For low-affinity binding (plots A to D), the value obtained for $K_{\mathrm{SV}}$ is essentially independent on the fitting procedure because in this case binding of the ligand to the protein does significantly deplete it in the aqueous phase. As the quenching efficiency decreases, the $K_{\mathrm{SV}}$ obtained decreases as well, underestimating $K_{b}$ if $s_{1}/s_{0}$ is assumed to be zero. For intermediate binding affinity (plots E to H), significant depletion of solute in the aqueous medium is observed and underestimates of $K_{b}$ will be obtained if $\left[L_{T}\right]$ is considered (black). For the other fitting procedures, the conclusions are the same as given above for low-affinity binding.

The case of high binding affinity is represented in plots I to L (and M to P with a logarithmic scale for the x axis). Neglect of ligand depletion leads to underestimation of $K_{b}$ by several orders of magnitude (black). Considering $\left[L_{W}\right]$ obtained from the best fit and the incorrect assumption that $s_{1}/s_{0}$= 0 (red) will also lead to underestimation of $K_{b}$ (for $s_{1}/s_{0}$ = 0.8, the value obtained for $K_{\mathrm{SV}}$ is 2.7 × 10^5^ M^−1^, leading to the same value for $K_{b}$ if full quenching is assumed). If both $\left[L_{W}\right]$ from the fit and the correct $s_{1}/s_{0}$ are considered (blue), $K_{\mathrm{SV}}$ is 1.5 × 10^6^ M^−1^ leading to $K_{b}$ = 7.5 × 10^6^ M^−1^, which although closer still underestimates the true value.

The correct parameters ($K_{b}$ and $s_{1}/s_{0}$) are obtained when the full variation of $S_{0}/S$ is analysed with Equation (32) (green) for all binding affinities and quenching efficiencies. However, some care must be taken regarding the sensitivity of the best fit to the estimated parameters. This is illustrated in Appendix A for low, intermediate and high affinity. At the concentration of protein considered in those simulations (1 µM), the sensitivity is high for low binding affinity, with the best fit, considering ratios of 1.5 or 0.67 between the estimated and the true $K_{b}$, leading to clear misfits (dark blue and orange). A good sensitivity is still obtained for the case of intermediate affinity, although when the x axis is represented in the logarithmic scale it is seen that a clear misfit is only obtained for ratios of 2 and 0.5. This is due to the significant ligand depletion, which is dependent on the estimated $K_{b}$ and shifts the curve of $S_{0}/S$ in the x axis. This effect becomes more evident for high-affinity binding (right plots), where essentially the same quality is obtained for the best fit with the estimated parameter varying from 0.5 to 2 times the true $K_{b}.$ A clear misfit of the experimental $S_{0}/S$ is only observed for an underestimation of $K_{b}$ by at least one order of magnitude, and essentially no sensitivity is observed for overestimation of $K_{b}$. A better sensitivity in the estimation of $K_{b}$ is obtained if the total concentration of protein is decreased, with $\left[P_{T}\right]=10^{-8}\;M$ leading to a good parameter estimation when $K_{b}=10^{8}\;M^{-1}$.

In conclusion, quenching of protein fluorescence due to ligand binding is an appropriate methodology to quantitatively characterize the binding to single binding site proteins, providing both the binding affinity and the quenching efficiency. However, the range of ligand concentrations must be large enough to lead to non-linear Stern–Volmer plots, and the whole variation of $S_{0}/S$ should be analysed with Equation (32). Additionally, to ensure a good sensitivity regarding the estimated binding affinity, the total concentration of protein should be lower than the estimated value for the dissociation constant. If the solubility of the ligand in the aqueous medium does not allow for reaching high enough concentrations to generate non-linear Stern–Volmer plots, the variation of $S_{0}/S$ must be analysed with Equation (22), and $K_{b}$ assumed equal to $K_{\mathrm{SV}}$. This procedure will underestimate $K_{b}$ if the quenching efficiency is not 100%. If $s_{1}/s_{0}$ may be obtained independently, an accurate estimation of $K_{b}$ may be obtained from $K_{\mathrm{SV}}$ using Equation (31).

When the proteins have several binding sites, the quenching of protein fluorescence may very easily lead to incorrect estimations of the binding parameters. To increase the accuracy, a wide range of ligand concentrations should be used to lead to non-linear Stern–Volmer plots (which in this case are expected to show upward curvature) and $S_{0}/S$ should be analysed with models considering several equal and independent binding sites, Equations (21) and (30). This procedure is expected to yield good estimates of the total binding affinity. If moderate-to-high binding affinity is obtained, the experiments should be repeated at a lower total concentration of protein to increase the confidence in the parameter estimated for high binding affinity.

In any case, the use of the simple Stern–Volmer equation with $\left[L_{W}\right]$ replaced by $\left[L_{T}\right]$ is an incorrect procedure that may underestimate the binding affinity by several orders of magnitude. The concentration of free ligand is dependent on the binding affinity and must be calculated for each set of parameters. The assumption that all binding sites in the protein are equal and independent leads to a simple procedure for the calculation of $\left[L_{W}\right]$, being a good compromise between accuracy and convenience.

### 2.4. Characterization of the Binding Parameters from Ligand Saturation Curves

It is sometimes easier or more convenient to follow the interaction of the ligand with the protein through a variation in the properties of the former. This is a convenient situation in the case of fluorescent ligands that change their fluorescence properties upon binding. It may also be a necessity if the ligand solubility does not allow attainment of protein saturation with ligand, limiting the maximum concentration achieved and/or leading to the formation of aggregates.

If the protein has only one binding site and the concentration of protein is known, the formalism is simple, and following protein saturation or ligand saturation leads to equivalent and accurate results. Equation (3) or Equation (7) can be used for this purpose, the latter equation only if the protein is in large excess. However, accuracy and equivalence are unwarranted when the protein has several binding sites, and/or when its concentration is not well-characterized. Some specific examples will be analysed below.

#### 2.4.1. Ligand Binding to Proteins with Multiple Binding Sites

The ligand saturation curves obtained for the case of proteins with three independent binding sites are shown in Figure 11 for different binding affinities and distinct relations between the affinities for the distinct binding sites in the protein.

The first observation is that a good fit is always obtained when the fraction of bound ligand is described by a model that considers an adjustable number of equal and independent binding sites, as in Equation (3). This clearly shows that the shape of the ligand saturation curve does not allow to distinguish between equal or different binding sites in the protein. For intermediate and high-affinity binding, it is observed that the best fit with an adjustable number of binding sites leads to a better description of the ligand saturation curve than when *n* = 1 is imposed (continuous vs. dashed lines, respectively). The misfit for *n* = 1 is clear in the case of high-affinity binding, but for low or intermediate affinity it would not be possible to distinguish between the two situations due to the noise in the real data.

The bottom plots in Figure 11 show the best fit of the simplified Equation (7), which considers that the protein is in large excess. As expected, a good fit is obtained for the case of low affinity (binding occurs at protein concentrations of several hundred µM and the ligand concentration is 1 µM). A clear misfit is obtained for high-affinity binding, and a reasonable fit for intermediate affinity.

The next question is whether the good fits lead to accurate estimations of the ligand overall binding affinity of the protein. Because different binding sites cannot be distinguished, the relevant properties are the overall binding affinity ($nK_{b}$ vs. $\sum K_{b}^{i}$), the highest binding affinity ($K_{b}$ vs. $K_{b}^{I}$), and the average binding affinity ($K_{b}$ vs. $\bar{K_{b}^{i}}=\sqrt[{n}]{\left(\prod_{i=1}^{n}K_{b}^{i}\right)}$). Because the sites are all assumed equal in the models considered to describe the ligand saturation profiles, the estimates obtained for the highest affinity and for the average affinity deviate significantly from the true values when the binding sites are different. However, the overall binding affinity may be estimated with good accuracy for most conditions. The estimates obtained from the best fit of ligand saturation curves for a large set of conditions are presented in Figure 12.

The fitting approach that yields the best estimate of the overall protein binding affinity is the use of the full equation considering an adjustable number of equal binding sites (plot A). This procedure leads to an excellent prediction of this property for low and intermediate affinities (less than 15% deviation) and reasonable predictions for high-affinity binding. In this case, the quality of the estimate gets poorer for higher ligand concentrations and for binding sites with similar (but not equal) affinities. This is due to a significant population of binding sites with intermediate affinity, which is not captured by the model that considers only one type of binding sites. When the number of binding sites is fixed at *n* = 1 (plot B) the predictions are usually good for low-, intermediate- and high-affinity binding at very low ligand concentrations (symbols in green), but overestimates the binding affinity by several orders of magnitude for high ligand concentrations (symbols in red). It should be noted that at high ligand concentrations the quality of the fit was very poor (Figure 11 plot C, dashed lines), which is related to the inadequate calculation of the free ligand concentration, as discussed when interpreting Figure 7.

The bottom plots in Figure 12 correspond to the estimates when the best fit of the fraction of ligand bound is obtained with the simplified equations that assume excess protein, Equation (5). As expected, the quality of the estimates is not dependent on the number of binding sites, as the results in plot C are equal to those in plot D, although the best fit was obtained for *n* ≠ 1 in plot C. This is because in the used equation *n* is always associated with $K_{b}$. The parameters obtained are only good in the case of low affinity binding, and lead to an underestimation of the overall binding affinity for intermediate and high-affinity binding. Surprisingly at first, the quality of the estimates improves as the concentration of ligand decreases, changing from $nK_{b}/\sum K_{b}^{i}$ equal to 0.03 at $\left[L_{T}\right]$ = 1 µM to 0.23 at $\left[L_{T}\right]$ = 0.1 µM in the case of $K_{b}^{I}=K_{b}^{\mathrm{II}}=K_{b}^{\mathrm{III}}$ = 10^8^ M^−1^. This is because, at very low ligand concentrations, significant ligand binding occurs only at total protein concentrations that are in fact much higher than the ligand concentration, as assumed in the derivation of the simplified solution.

From this detailed analysis it may be concluded that, when analysing ligand saturation without knowledge of the detailed binding details (number of binding sites and whether they are equal or different), it is a reasonable approach to consider that all binding sites are equal, and perform the analysis in terms of binding sites using the full equation. If a good fit is obtained, this procedure is expected to yield an accurate prediction of the overall protein-binding affinity. The correct number of binding sites may also be obtained if they are all equal and if binding leads to a significant depletion of the total protein. However, if the binding sites are not equal, the value of *n* estimated from the best fit is an underestimation. The deviation between the estimated and the real number of binding sites in the protein depends on the values of the distinct binding affinities, the relation being $n\;K_{b}=\sum K_{b}^{i}$ when a good estimate of the overall protein-binding affinity is obtained. Concomitantly, following the ligand saturation does not lead to good estimates of the highest binding affinity. Exceptions occur when the affinity is the same for all binding sites, or when the difference between the distinct binding sites is very large (more than one order of magnitude), in this case leading only to the characterization of the highest affinity binding site.

There are several examples in the literature where this procedure was followed to characterize the association of fluorescent amphiphiles with proteins [29,52,57,58,59,60,73,74,75,76,77]. In most cases the objective was only to evaluate the overall binding affinity. As explained above, this parameter is obtained accurately, even for the case of proteins with multiple binding sites when very low concentrations of ligands are used [29,58,59,60,73,74].

#### 2.4.2. Ligand Binding as a Partition Coefficient

In some situations, the protein under study is not well-characterized (e.g., in terms of protein molecular weight, and number of binding sites), and only the total mass of protein in solution is known. In these cases, it is useful to describe binding as a partition coefficient between the aqueous medium and the protein. It may also be convenient to describe binding in terms of partition for the case of proteins with several binding sites such as serum albumin [72,78,79]. If the concentration of the ligand is much lower than that of the protein, the volume of protein available for binding ($V_{P}$) is maintained throughout the titration, and the equilibrium distribution of ligand is given by Equation (33).
(33)$$L_{W}\overset{K_{P}^{W\to P}\;V_{P}}{↔}L_{P}\;;\;K_{P}^{W\to P}=\frac{nL_{P}/V_{P}}{nL_{W}/V_{W}}\;$$

The partition coefficient obtained is related with the overall equilibrium association by:(34)$$\begin{array}{ll} K_{P}^{W\to P} & =\frac{nL_{P}}{nL_{W}}\frac{V_{w}}{n_{P}\bar{V_{P}}}≅\frac{nL_{P}}{nL_{W}}\frac{V_{T}}{n_{P}\bar{V_{P}}}=\frac{nL_{P}}{nL_{W}}\frac{1}{\left[P\right]}\frac{1}{\bar{V_{P}}}=K_{b}^{\mathrm{obs}}\frac{1}{\bar{V_{P}}}\;\;\\ & \Leftrightarrow \;\;K_{b}^{\mathrm{obs}}=K_{P}^{W\to P}\bar{V_{P}} \end{array}$$
where $K_{b}^{\mathrm{obs}}$ is the observed association constant when considering the whole protein, and $\bar{V_{P}}$ is the molar volume of the whole protein. The assumption made (*V*_w_ ≅ *V*_T_) is valid for dilute solutions, which is the case for studies performed using model systems. Its validity must be verified if this formalism is applied to biological systems or when very high concentrations of binding agents are used.

The concentrations of ligand associated with the protein and in the aqueous phase are given by:(35)$$\begin{array}{l} \left[L_{P}\right]=\left[L_{T}\right]\frac{K_{P}^{W\to P}\left(V_{P}/V_{T}\right)}{V_{W}/V_{T}+K_{P}^{W\to P}\left(V_{P}/V_{T}\right)}\\ \left[L_{W}\right]=\left[L_{T}\right]\frac{V_{W}/V_{T}}{V_{W}/V_{T}+K_{P}^{W\to P}\left(V_{P}/V_{T}\right)} \end{array}$$

If the concentration of ligand is comparable to that of the protein, it is not correct to assume that the volume of the protein available for binding remains constant throughout the titration. In this case, it is necessary to consider the available volume of protein, $V_{P}^{*}$, which is given by:(36)$$V_{P}^{*}=V_{P}-nL_{P}\;\frac{\bar{V_{P}}}{\#}$$
where *#* is the number of binding sites per molar volume. The scheme describing the partition of the ligand between the aqueous medium and the protein is obtained replacing $V_{P}^{}$ by $V_{P}^{*}$ in Equation (33).

The volume of protein available to interact with the ligand is calculated from the quadratic equation:(37)$${\left(\frac{V_{P}^{*}}{V_{T}}\right)}^{2}K_{P}^{W\to P}+\frac{V_{P}^{*}}{V_{T}}\left\{\frac{V_{w}}{V_{T}}+K_{P}^{W\to P}\left(\frac{nS_{T}}{V_{T}}\frac{\bar{V_{P}}}{\#}-\frac{V_{P}}{V_{T}}\right)\right\}-\frac{V_{w}}{V_{T}}\frac{V_{P}}{V_{T}}=0$$
where the physically meaningful solution is always ${\left(V_{P}^{*}\right)}_{+}$.

The concentration of ligand associated with the protein or in the aqueous phase may be calculated from the equivalent to Equation (35) but with the volume of the protein being replaced by $V_{P}^{*}$, which is given by the solution of Equation (37).

The application of this formalism to the analysis of ligand binding to proteins with multiple binding sites leads to results identical to those shown in Figure 11 and Figure 12. A good fit is always obtained when considering a variable number of binding sites, and the volume of free protein from Equation (37). The partition coefficient is a good estimate for the overall binding affinity, with $K_{b}^{\mathrm{obs}}$ being close to $\sum K_{b}^{i}$. The simplified Equation (35) is only valid for very low ligand concentrations or very small $K_{P}^{W\to P}$, which in both cases leads to a low saturation of the protein.

The advantage of this formalism is that it allows to recover the overall binding affinity even when the concentration of protein is not known, only its total mass or volume. If the ligand concentration is high enough to saturate the protein, the best fit is dependent on n, and an estimate is obtained for the number of binding sites in the protein (considered equal and independent). From this it is possible to estimate the volume per binding site. However, if the best fit is not dependent on n, only the partition coefficient is obtained, and it is not possible to calculate the observed binding constant. Nevertheless, from the partition coefficient one may directly calculate the fraction of ligand associated with the protein for a given amount of protein, which is usually the relevant property.

Another advantage of this formalism is the ease of its application to the description of ligand distribution in heterogeneous media with increasing complexity. An example is when membranes and proteins are present in the same aqueous solution, as observed in all biological systems. This situation will be considered in the next section.

## 3. Ligand Distribution in the Presence of Proteins and Membranes

Ligand association with membranes is usually treated as a partition because the membranes do not have well-defined binding sites and because they can accumulate relatively high amounts of ligand without changing their binding properties. For small uncharged or singly charged ligands, the membrane may accommodate at least up to 5 mol% of the former without changing their binding properties [33,34]; depending on the ligand and membrane composition, local ligand concentrations as high as 20 mol% may be achieved without significant changes in the membrane binding affinity [80,81,82]. Considering the typical molar volume of a membrane lipid (0.8 dm^3^/mol), the local ligand concentration in the membrane may thus exceed 0.1 M.

The use of the partition formalism allows a direct comparison of the binding affinity of lipid membranes and proteins. For a protein with $\bar{V_{P}}$ = 50 dm^3^/mol (corresponding to a molecular weight of ~70 kg/mol), a moderate binding affinity $K_{b}^{\mathrm{obs}}$ = 10^6^ M^−1^ corresponds to $K_{P}^{W\to P}$ equal to 2 × 10^4^. At low ligand concentrations, for an equivalent volume of protein and lipid bilayer the ligand will be equally distributed for $K_{P}^{W\to \mathrm{Lb}}$ equal to 2 × 10^4^. However, at high ligand concentrations the protein becomes saturated while the membrane is still able to accommodate the ligand. This will shift the distribution of ligand from the protein to the membrane. The maximum local concentration of ligand in the protein is equal to $1/\left(\bar{V_{P}}/\#\right).$ For a protein with $\bar{V_{P}}$ = 50 dm^3^/mol and a single binding site this concentration is 0.02 M, and is thus lower than the maximum ligand concentration achievable in a membrane without changing its properties as a binding agent.

A quantitative analysis of the ligand distribution between the aqueous medium, the protein and the lipid membrane will be presented in the following sections. Focus will be given to the effect of ligand concentrations regarding the fraction of ligand bound to the protein and membrane. Several distinct conditions will be considered: proteins soluble in the aqueous phase (case I), and membrane proteins (case II to IV), as shown in Figure 13.

### 3.1. Lipid Membranes and Proteins in Solution

When using the partition formalism, the equations that describe the distribution of ligand between the aqueous phase and the different binding agents are easily obtained. This distribution is given by Equation (38) for the case of lipid membranes and aqueous soluble proteins (case I).
(38)$$\begin{array}{l} \left[L_{W}\right]=\left[L_{T}\right]\frac{V_{W}/V_{T}}{V_{W}/V_{T}+K_{P}^{W\to P}\left(V_{P}^{*}/V_{T}\right)+K_{P}^{W\to \mathrm{Lb}}\left(V_{\mathrm{Lb}}/V_{T}\right)}\\ \left[L_{P}\right]=\left[L_{T}\right]\frac{K_{P}^{W\to P}\left(V_{P}^{*}/V_{T}\right)}{V_{W}/V_{T}+K_{P}^{W\to P}\left(V_{P}^{*}/V_{T}\right)+K_{P}^{W\to \mathrm{Lb}}\left(V_{\mathrm{Lb}}/V_{T}\right)}\\ \left[L_{\mathrm{Lb}}\right]=\left[L_{T}\right]\frac{K_{P}^{W\to \mathrm{Lb}}\left(V_{\mathrm{Lb}}/V_{T}\right)}{V_{W}/V_{T}+K_{P}^{W\to P}\left(V_{P}^{*}/V_{T}\right)+K_{P}^{W\to \mathrm{Lb}}\left(V_{\mathrm{Lb}}/V_{T}\right)} \end{array}$$

The volume of lipid bilayer available to receive ligand is considered constant (non-saturable binding agent), while that of the protein depends on the amount of ligand bound. The volume of protein available to the ligand is obtained from the solution of the quadratic equation given in Equation (39), the physically meaningful solution being always $x_{+}$:(39)$${\left(\frac{V_{P}^{*}}{V_{T}}\right)}^{2}K_{P}^{W\to P}+\frac{V_{P}^{*}}{V_{T}}\left\{\frac{V_{w}}{V_{T}}+K_{P}^{W\to \mathrm{Lb}}\frac{V_{\mathrm{Lb}}}{V_{T}}+K_{P}^{W\to P}\left(\frac{nS_{T}}{V_{T}}\frac{\bar{V_{P}}}{\#}-\frac{V_{P}}{V_{T}}\right)\right\}-\frac{V_{P}}{V_{T}}\left(\frac{V_{w}}{V_{T}}+K_{P}^{W\to \mathrm{Lb}}\frac{V_{\mathrm{Lb}}}{V_{T}}\right)=0$$

The effect of ligand concentration on the membrane volume and that of protein available for ligand binding is represented in Figure 14A for the case of moderate affinity of the ligand for the protein ($K_{b}^{\mathrm{obs}}$= 5 × 10^5^ M^−1^, $K_{P}^{W\to P}$ = 10^4^), and low affinity for the lipid bilayer ($K_{P}^{W\to \mathrm{Lb}}$= 10^2^). In the absence of ligand, at the protein and lipid concentrations considered, the volume of protein available for ligand binding is slightly higher than that of the lipid membrane. However, as the concentration of ligand is increased, the protein binding sites are occupied and the protein volume available for binding decreases. In contrast, the membrane volume available for binding is maintained. The fraction of ligand in the distinct media is presented in plot B. At very low ligand concentrations, most ligand is associated with the protein. However, as the protein becomes saturated with the ligand, it accumulates in the aqueous medium and in the membrane. For the low membrane affinity considered in this simulation, only about 10% of the ligand is associated with the membrane. A higher membrane affinity is considered in plots C ($K_{P}^{W\to \mathrm{Lb}}$ = 10^3^) and D ($K_{P}^{W\to \mathrm{Lb}}$ = 10^4^). In these conditions the fraction of ligand associated with the membrane becomes significant and may represent up to 90%. It should be noted that in plot D, the overall affinities of the ligand for the protein and membrane are equal $\left(K_{P}^{W\to P}=K_{P}^{W\to \mathrm{Lb}}\right)$, and that in the absence of ligand the volume of protein available is slightly higher than that of the membrane. However, the membrane becomes the dominant binding agent for ligand concentrations higher than 9 µM.

### 3.2. Membranes Containing Lipids and Membrane Proteins

In this case one has to consider three possibilities for the equilibration of ligand with the membrane protein: from the aqueous phase only (Case II), from the membrane only (Case III), or from both (Case IV).

The equations that describe the distribution of ligand at equilibrium for case II are the same as those for soluble proteins, Equations (38) and (39). All the observations from the previous section are thus valid in this case, with the presence of the membrane decreasing the apparent affinity of the ligand for the protein.

If ligand binding to the protein occurs only after ligand association with the membrane (Case III), the equilibrium distribution of the ligand may be calculated from Equations (40) and (41).
(40)$$\begin{array}{l} \left[L_{W}\right]=\left[L_{T}\right]\frac{V_{W}/V_{T}}{V_{W}/V_{T}+K_{P}^{W\to \mathrm{Lb}}\left\{\left(V_{\mathrm{Lb}}/V_{T}\right)+\left(V_{P}^{*}/V_{T}\right)K_{P}^{\mathrm{Lb}\to P}\right\}}\\ \left[L_{P}\right]=\left[L_{T}\right]\frac{K_{P}^{W\to \mathrm{Lb}}K_{P}^{\mathrm{Lb}\to P}\left(V_{P}^{*}/V_{T}\right)}{V_{W}/V_{T}+K_{P}^{W\to \mathrm{Lb}}\left\{\left(V_{\mathrm{Lb}}/V_{T}\right)+\left(V_{P}^{*}/V_{T}\right)K_{P}^{\mathrm{Lb}\to P}\right\}}\\ \left[L_{\mathrm{Lb}}\right]=\left[L_{T}\right]\frac{K_{P}^{W\to \mathrm{Lb}}\left(V_{\mathrm{Lb}}/V_{T}\right)}{V_{W}/V_{T}+K_{P}^{W\to \mathrm{Lb}}\left\{\left(V_{\mathrm{Lb}}/V_{T}\right)+\left(V_{P}^{*}/V_{T}\right)K_{P}^{\mathrm{Lb}\to P}\right\}} \end{array}$$
(41)$${\left(\frac{V_{P}^{*}}{V_{T}}\right)}^{2}K_{P}^{W\to \mathrm{Lb}}K_{P}^{\mathrm{Lb}\to P}+\frac{V_{P}^{*}}{V_{T}}\left\{\frac{V_{w}}{V_{T}}+K_{P}^{W\to \mathrm{Lb}}\frac{V_{\mathrm{Lb}}}{V_{T}}+K_{P}^{W\to \mathrm{Lb}}K_{P}^{\mathrm{Lb}\to P}\left(\frac{nS_{T}}{V_{T}}\frac{\bar{V_{P}}}{\#}-\frac{V_{P}}{V_{T}}\right)\right\}-\frac{V_{P}}{V_{T}}\left(\frac{V_{w}}{V_{T}}+K_{P}^{W\to \mathrm{Lb}}\frac{V_{\mathrm{Lb}}}{V_{T}}\right)=0$$

The results obtained for a moderate relative affinity between the membrane and the protein $\left(K_{P}^{\mathrm{Lb}\to P}=10\right)$ are shown in Figure 15, at affinities between the aqueous medium and the membrane, $K_{P}^{W\to \mathrm{Lb}}$, increasing from 10^2^ (plot A) to 10^4^ (plot C). In contrast to what is observed for proteins in the aqueous medium (Figure 14), increasing the affinity of the ligand for the lipid bilayer leads to an apparent increase in the ligand affinity for the protein. This result was expected because it is the ligand in the membrane that is able to interact with the protein. However, a closer analysis of Figure 15 shows that this increased affinity for the protein is only significant for low ligand concentrations. As the protein-binding sites become saturated, most ligand accumulates in the membrane despite the lower intrinsic affinity.

Even though this model assumes that the ligand cannot associate with the membrane protein directly from the aqueous medium, this step closes a thermodynamic circle and thus the apparent partition coefficient may be calculated from the other two equilibria, Equation (42).
(42)$$K_{P}^{W\to P}=K_{P}^{W\to \mathrm{Lb}}K_{P}^{\mathrm{Lb}\to P}$$

Due to the thermodynamic cycle involving the ligand in the aqueous medium and associated with the lipid bilayer or with the protein, Case IV in Figure 13 cannot be formally distinguished from Case III in what concerns the equilibrium distribution of the ligand. In fact, Case II is also formally indistinguishable from Case III and IV, Equations (40) and (41) being obtained if $K_{P}^{W\to P}$ in Equations (38) and (39) is replaced by its relation with $K_{P}^{W\to \mathrm{Lb}}$ and $K_{P}^{\mathrm{Lb}\to P}$ given by Equation (42). Another mechanism could additionally be considered, with the ligand interaction with the lipid bilayer being mediated by interaction with the membrane protein, which is also described by the same formalism. This situation was not explicitly considered because interaction with the lipid membrane is usually non-specific, being observed for the vast majority of ligands with moderate and high hydrophobicity.

The formalism developed in this section may be of particular relevance for the characterization of ligand association with membrane proteins. It provides a simple method to quantitatively compare the ligand affinity for the lipid bilayer and for the proteins, and to evaluate the effect of ligand concentration and relative concentration of the membrane proteins. Those aspects will be further explored in a dedicated manuscript. For now, we will just define another important experimental observable, the partition coefficient between the aqueous medium and the membrane as a whole, as shown in Equation (43).
(43)$$K_{P}^{W\to M}=\frac{\left(nL_{\mathrm{Lb}}+nL_{P}\right)/\left(V_{\mathrm{Lb}}+V_{P}\right)}{nL_{W}/V_{W}}=\frac{K_{P}^{W\to \mathrm{Lb}}V_{\mathrm{Lb}}+K_{P}^{W\to P}V_{P}}{V_{\mathrm{Lb}}+V_{P}}=\frac{K_{P}^{W\to \mathrm{Lb}}\left(V_{\mathrm{Lb}}+K_{P}^{\mathrm{Lb}\to P}V_{P}\right)}{V_{\mathrm{Lb}}+V_{P}}$$

Thus, the characterization of the partition coefficient at low ligand concentrations in the presence and absence of the membrane protein, allows obtaining the water to protein and the lipid bilayer to protein partition coefficients. This may be of high relevance when studying interactions with biological membranes.

The situations discussed above correspond to different mechanisms of signal transduction and ligand transport through biological membranes with impact in the areas of biochemistry, pharmacology, neurochemistry and endocrinology. Distinction between the mechanisms of interaction with the membrane protein has been attempted, namely on the basis of the dependence on ligand hydrophobicity [83,84]. However, an increase in ligand lipophilicity would lead to an increased apparent affinity for the membrane protein in all cases, provided that the relative affinity for the lipid bilayer and the protein is maintained. The mechanisms may eventually be distinguished if the kinetics of the interaction is accessed. If the ligand in the aqueous medium can only interact with the membrane, a relative accumulation of ligand in the membrane is expected at short times followed by its redistribution to the membrane protein. The reverse is expected if the ligand in the aqueous medium can only interact with the membrane protein. If direct interaction is established with both binding agents, the time evolution in each medium would depend on the relative rate and equilibrium constants.

## Figures and Tables

**Figure 1 ijms-23-09757-f001:**
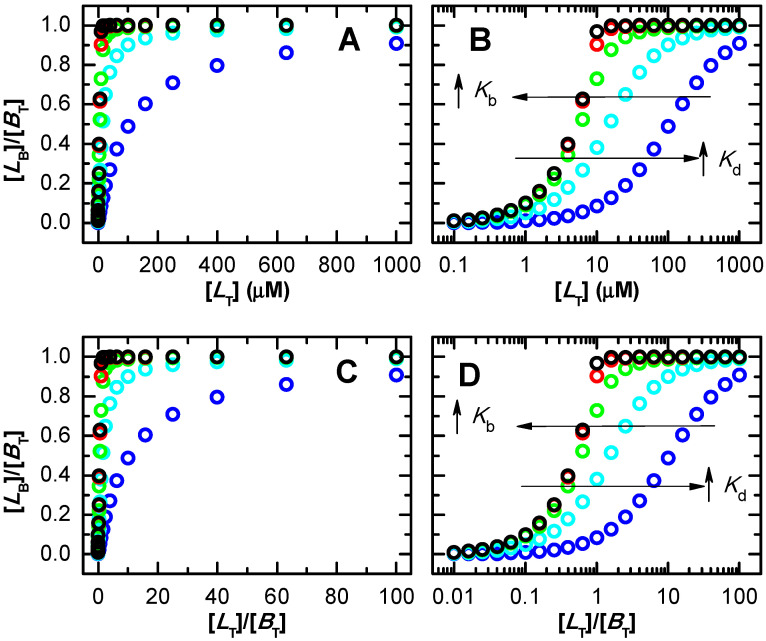
Saturation of binding sites predicted from the full description of the system, Equation (3), as a function of the total concentration of ligand (plots (**A**,**B**)), or of the ratio of total ligand to binding sites (plots (**C**,**D**)). Note the logarithmic scale on the x-axis of plots (**B**,**D**). The total concentration of binding sites was 10 µM, and the association affinity ($K_{d}$) was varied: 100 µM (**◯**), 10 µM (**◯**), 1 µM (**◯**), 0.1 µM (**◯**), and 0.01 µM (**◯**).

**Figure 2 ijms-23-09757-f002:**
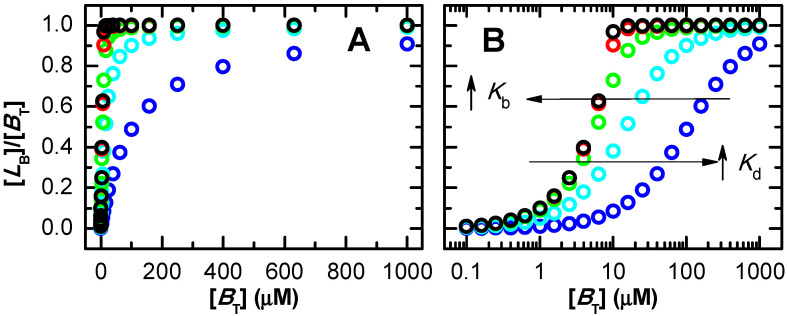
Fraction of bound ligand predicted from the full description of the system, Equation (3), as a function of the total concentration of binding sites, represented with linear (**A**) or logarithmic (**B**) x-axis scale. The total concentration of ligand was 10 µM, and the association affinity was varied from $K_{d}$ equal to 100 µM (◯), 10 µM (**◯**), 1 µM (**◯**), 0.1 µM (**◯**), and 0.01 µM (**◯**).

**Figure 3 ijms-23-09757-f003:**
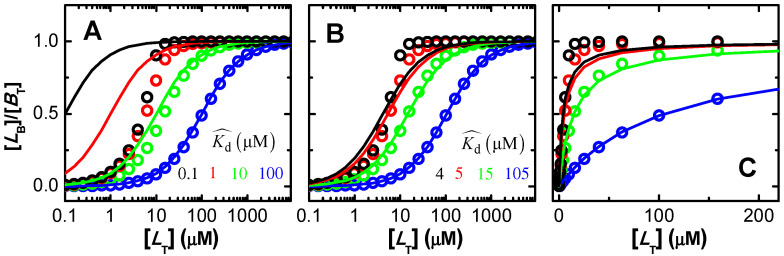
Saturation of binding sites predicted from the full description of the system (Equation (3), circles) for $\left[B_{T}\right]$ = 10 µM and different dissociation constants ($K_{d}$ ): 100 μM (◯), 10 µM (**◯**), 1 µM (**◯**), and 0.1 µM (**◯**). The lines were calculated from Equation (6) (excess ligand approximation). Plot (**A**)—predicted saturation of binding sites for the different binding affinities considered; Plots (**B**,**C**)—best fit of Equation (6) to the saturation of binding sites observed (note the logarithmic scale in plot (**B**) and linear scale in plot (**C**)). The dissociation constants obtained from the best fit of Equation (6) are indicated in plot (**B**).

**Figure 4 ijms-23-09757-f004:**
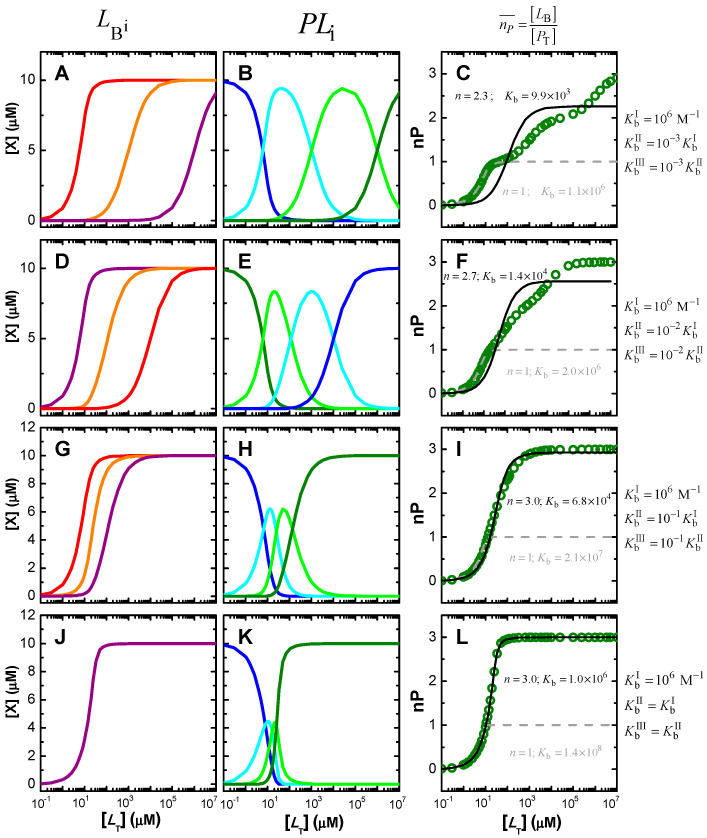
Variation of the concentration of occupied binding sites ($LB^{I}$
**―**, $LB^{\mathrm{II}}$
**―**, and $LB^{\mathrm{III}}$ ―) (**A**,**D**,**G**,**J**), protein with *i* ligands bound (P
**―**, $PL_{1}$
**―**, $PL_{2}$
―, and $PL_{3}$
―) (**B**,**E**,**H**,**K**), and average number of ligands per protein ($\bar{n_{P}}$
**◯**) (**C**,**F**,**I**,**L**), as a function of $\left[L_{T}\right]$ for $\left[P_{T}\right]$ = 10 µM. The microscopic association constant of highest affinity was equal to 10^6^ M^−1^, while the relative values of the intermediate and lower affinity binding sites were varied from top to bottom. The lines in the right plots (**C**,**F**,**I**,**L**) are the best fits considering only one type of binding sites, Equation (3), with the number of binding sites fixed at 1 (**- - -**) or as an adjustable variable (**―**), and the estimated values shown in the respective plot.

**Figure 5 ijms-23-09757-f005:**
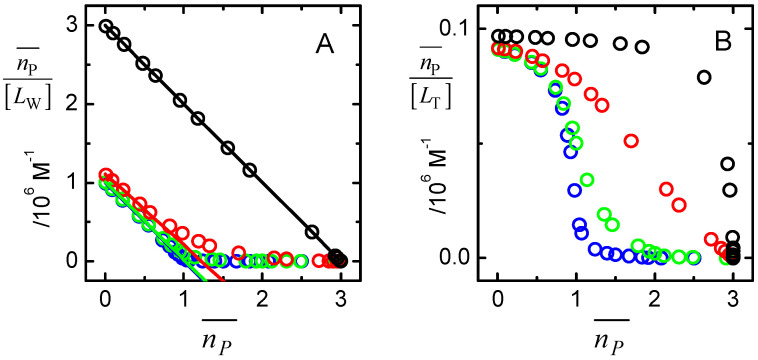
Scatchard plots for the simulation results considering ligand association to proteins with 3 independent binding sites with the microscopic association constant of highest affinity $K_{b}^{I}$ = 10^6^ M^−1^, and different relative values of the intermediate and lower affinity binding sites: $K_{b}^{\mathrm{III}}=K_{b}^{\mathrm{II}}=K_{b}^{I}$ (◯), $K_{b}^{\mathrm{III}}=10^{-1}K_{b}^{\mathrm{II}}\;;\;K_{b}^{\mathrm{II}}=10^{-1}K_{b}^{I}$ (◯), $K_{b}^{\mathrm{III}}=10^{-2}K_{b}^{\mathrm{II}}\;;\;K_{b}^{\mathrm{II}}=10^{-2}K_{b}^{I}$ (◯), $K_{b}^{\mathrm{III}}=10^{-3}K_{b}^{\mathrm{II}}\;;\;K_{b}^{\mathrm{II}}=10^{-3}K_{b}^{I}$ (◯). The lines in plot A are best fits of straight lines to the region of low protein saturation. Note that in plot (**A**) the protein saturation in the y-axis is divided by the concentration of free ligand, while in plot (**B**) the total ligand concentration is used.

**Figure 6 ijms-23-09757-f006:**
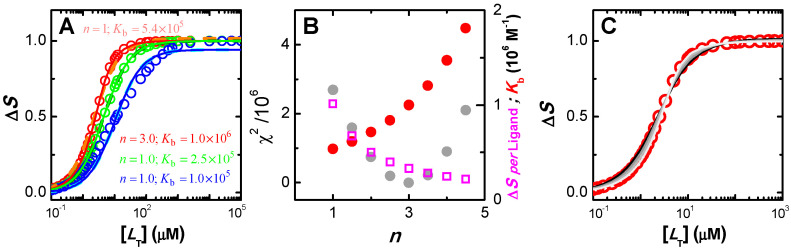
Plot (**A**): variation of a hypothetical signal proportional to the number of ligands associated with a protein containing 3 independent binding sites:$K_{b}^{I}$ = 10^6^ M^−1^; and $K_{b}^{I}/K_{b}^{\mathrm{II}}=K_{b}^{\mathrm{II}}/K_{b}^{\mathrm{III}}=10$ (◯), $K_{b}^{I}/K_{b}^{\mathrm{II}}=K_{b}^{\mathrm{II}}/K_{b}^{\mathrm{III}}=3$ (◯), and $K_{b}^{I}/K_{b}^{\mathrm{II}}=K_{b}^{\mathrm{II}}/K_{b}^{\mathrm{III}}=1$ (◯). The lines are the best fits considering only one type of binding sites, Equation (3), with the number of binding sites fixed at 1 (thick dashed lines) or as an adjustable variable (thin continuous lines). The parameters obtained from the best fit are also shown. Plot (**B**): effect of the number of binding sites per protein in terms of the quality of the best fit (●), association constant (●) and signal variation per ligand bound (**☐**), for the case $K_{b}^{I}/K_{b}^{\mathrm{II}}=K_{b}^{\mathrm{II}}/K_{b}^{\mathrm{III}}=1$. Plot (**C**): Fitting curves obtained for *n* = 1 (☐), *n* = 2 (☐), *n* = 3 (☐), and *n* = 4 (☐).

**Figure 7 ijms-23-09757-f007:**
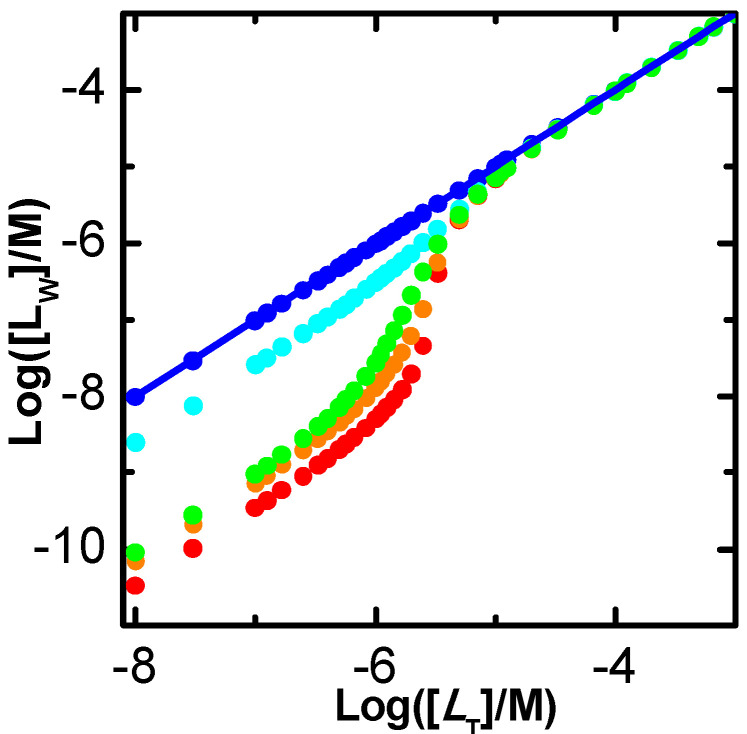
Variation of the concentration of free ligand in solution as a function of the total ligand concentration, for $\left[P_{T}\right]$ = 1 µM and considering 3 equal and independent binding sites with $K_{b}^{}$ equal to 10^4^ M^−1^ (●), 10^6^ M^−1^ (●) or 10^8^ M^-1^ (●), and for different binding sites with $K_{b}^{I}$ = 10^8^ M^−1^, and $K_{b}^{I}/K_{b}^{\mathrm{II}}=K_{b}^{\mathrm{II}}/K_{b}^{\mathrm{III}}=3$ (●), or $K_{b}^{I}/K_{b}^{\mathrm{II}}=K_{b}^{\mathrm{II}}/K_{b}^{\mathrm{III}}=10$ (●). The blue line is for $\left[L_{W}\right]=\left[L_{T}\right]$.

**Figure 8 ijms-23-09757-f008:**
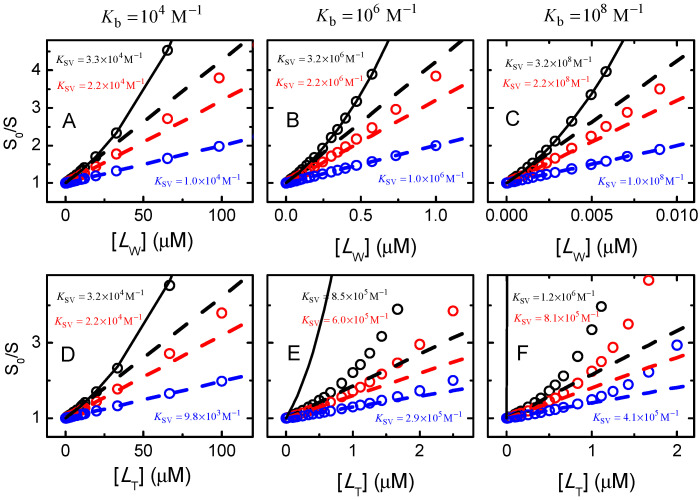
Stern–Volmer plots for the quenching of protein fluorescence due to ligand binding. Three equal and independent binding sites in the protein are considered, with low ($K_{b}^{}$ = 10^4^ M^−1^, plot (**A**,**D**)), intermediate ($K_{b}^{}$ = 10^6^ M^−1^, plot (**B**,**E**)) or high ($K_{b}^{}$ = 10^8^ M^−1^, plot (**C**,**F**)) affinity. 100% quenching of protein fluorescence is considered for the ligand associated with any of the binding sites ($s_{i}=0,\;i\neq 0$, **◯**), 40%, 70% or 100% for ligand in binding site I, II, or III respectively ($s_{I}=0.6,\;s_{\mathrm{II}}=0.3,\;s_{\mathrm{III}}=0$, **◯**), or no quenching by ligands in binding sites I or II, and 100% quenching by ligand in binding site III ($s_{I}=s_{\mathrm{II}}=1,\;s_{\mathrm{III}}=0$, **◯**). The values obtained for $K_{\mathrm{SV}}$ from the slope at $S_{0}/S\leq 1.1$ (Equation (22), dashed lines) are given in each plot with the corresponding colours. The continuous lines are the predictions from Equation (25), considering the “true” binding parameters. In the upper plots the free ligand concentration is considered, while the bottom plots consider the total concentration of ligand.

**Figure 9 ijms-23-09757-f009:**
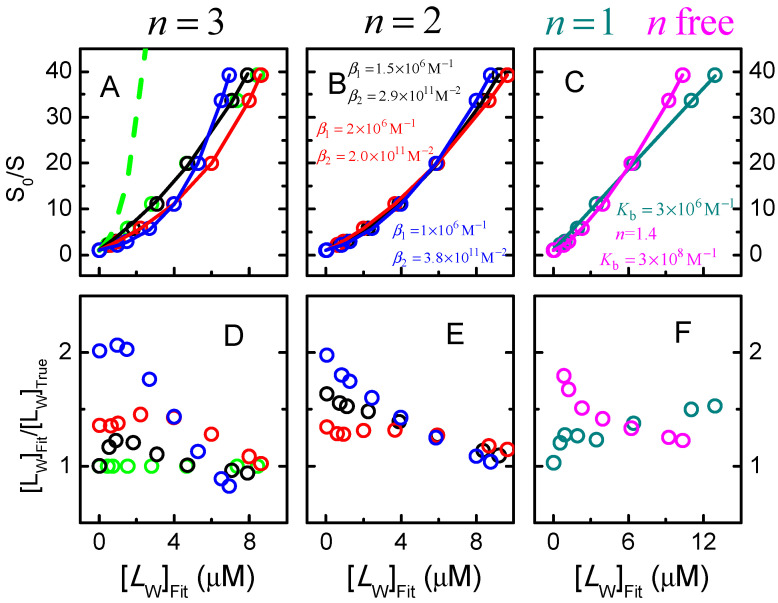
Stern–Volmer plots obtained for the best fit of the results simulated for ligand binding to a protein with 3 equal and independent binding sites ($K_{b}^{}$ = 10^6^ M^−1^; corresponding to $\beta_{1}$ = 3 × 10^6^ M^−1^, $\beta_{2}$ = 3 × 10^12^ M^−2^, and $\beta_{3}$ = 1 × 10^18^ M^−3^) and partial fluorescence quenching by the ligand in sites I and II ($s_{I}=0.6,\;s_{\mathrm{II}}=0.3,\;s_{\mathrm{III}}=0$ ). The upper plots show $S_{0}/S$ as a function of the concentration of free ligand obtained from the best fit ${\left[L_{W}\right]}_{\mathrm{FIT}}$, and the ratio of ${\left[L_{W}\right]}_{\mathrm{FIT}}$ and the “true” concentration of free ligand is shown in the lower plots. Plot (**A**) $S_{0}/S$ obtained for the binding and quenching parameters considered in the simulation (**◯**), and predicted by Equation (25) for the same values of $\beta_{i}$ (▪ ▪ ▪). Some examples of best fit obtained for different values of $\beta_{i}$ are also shown: $\beta_{1}$ = 3 × 10^6^ M^−1^ (fixed), leading to $\beta_{2}$ = 1.2 × 10^11^ M^−2^, and $\beta_{3}$ = 1.5 × 10^16^ M^−3^ (**◯** and ☐); $\beta_{1}$ = 2 × 10^6^ M^−1^ (fixed), leading to $\beta_{2}$ = 1.4 × 10^8^ M^−2^, and $\beta_{3}$ = 3.3 × 10^16^ M^−3^ (**◯** and ☐); and $\beta_{1}$ = 1 × 10^6^ M^−1^ (fixed), leading to $\beta_{2}$ = 2.3 × 10^8^ M^−2^, and $\beta_{3}$ = 9.4 × 10^16^ M^−3^ (**◯** and ☐). Several combinations of association constants considering 2 binding sites are shown in Plot (**B**), and considering 1 or an adjustable number of binding sites are shown in Plot (**C**). The resulting association constants are given in the plot with the same colour as that of the respective curves.

**Figure 10 ijms-23-09757-f010:**
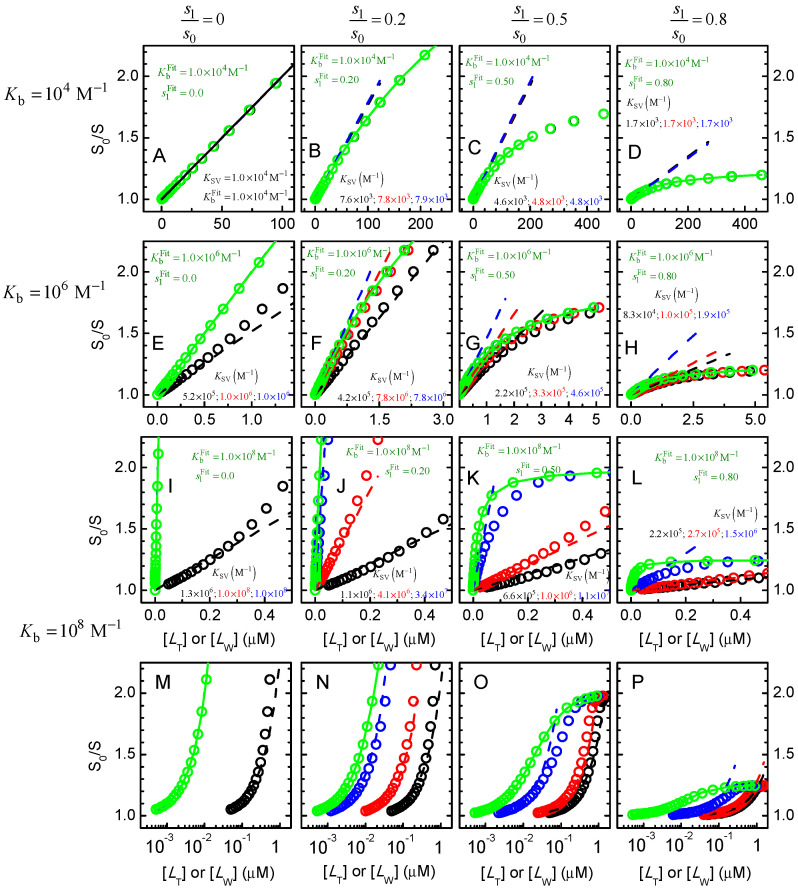
Stern–Volmer plots for the quenching of protein fluorescence (at $\left[P_{T}\right]=10^{-6}\;M$) due to ligand binding. One binding site in the protein is considered, with low ($K_{b}^{}$ = 10^4^ M^−1^, plot (**A**–**D**)), intermediate ($K_{b}^{}$ = 10^6^ M^−1^, plot (**E**–**H**)) or high ($K_{b}^{}$ = 10^8^ M^−1^, plot (**I**–**P**), note the logarithmic scale in the y axis for the lower plots) affinity. Different efficiencies of quenching of protein fluorescence are considered ($s_{1}/s_{0}=0$, leftmost plots; $s_{1}/s_{0}=0.2$, 2nd from left; $s_{1}/s_{0}=0.5$, 3rd from left; and $s_{1}/s_{0}=0.8$, rightmost). The $S_{0}/S$ ratio at low quenching was analysed with the simplified Stern–Volmer Equation (22) considering that $\left[L_{W}\right]=\left[L_{T}\right]$ (◯, and ▪ ▪ ▪), or calculating $\left[L_{W}\right]$ from Equation (1) and $x_{-}$ solution of (3) assuming that $K_{b}=K_{\mathrm{SV}}$ (◯, and ▪ ▪ ▪), or using Equation (31) to calculate $K_{b}$ from $K_{\mathrm{SV}}$ (◯, and ▪ ▪ ▪). The full variation of $S_{0}/S$ was also analysed with Equation (32), to obtain directly $K_{b}$ and $s_{1}/s_{0}$ (◯, and ▪ ▪ ▪). The values of $K_{\mathrm{SV}}$ obtained from the best fit at low $S_{0}/S$ are shown in the plots with the same colour as the respective best fit lines. The parameters obtained from the best fit of the whole variation of $S_{0}/S$ are shown in green.

**Figure 11 ijms-23-09757-f011:**
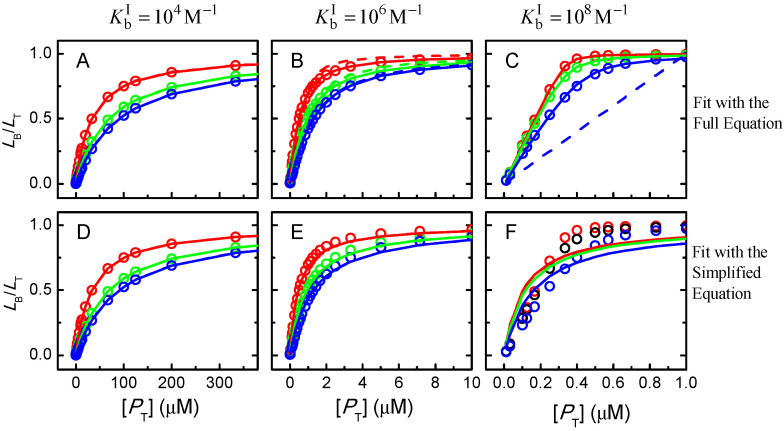
Fraction of ligand bound to a protein with 3 independent binding sites of equal affinity (**◯**), with $K_{b}^{I}/K_{b}^{\mathrm{II}}=K_{b}^{\mathrm{II}}/K_{b}^{\mathrm{III}}=3$ (**◯**) or with $K_{b}^{I}/K_{b}^{\mathrm{II}}=K_{b}^{\mathrm{II}}/K_{b}^{\mathrm{III}}=10$ (**◯**) for binding with low affinity ($K_{b}^{I}=10^{4}\;M^{-1}$, plots (**A**,**D**)), intermediate affinity ($K_{b}^{I}=10^{6}\;M^{-1}$, plots (**B**,**E**)), and high affinity ($K_{b}^{I}=10^{8}\;M^{-1}$, plots (**C**,**F**)). In the upper plots the lines are the best fit of Equation (3) with an adjustable number of binding sites *n* with equal affinity *K*_b_ (continuous lines), or for a fixed number of binding sites *n* = 1 (dashed lines). The equivalent is shown in the lower plots but showing the best fit of the simplified Equation (7), considering that the protein is in large excess. The total concentration of ligand is 1 µM in all plots.

**Figure 12 ijms-23-09757-f012:**
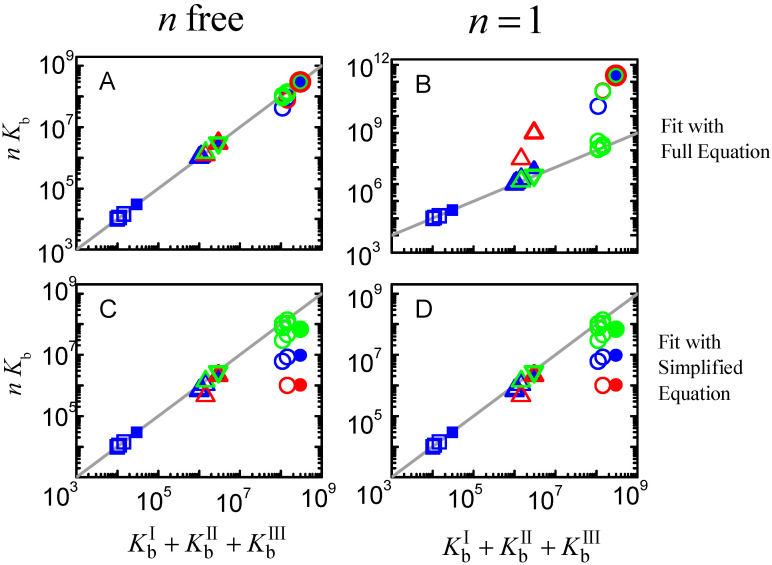
Relation between the overall binding affinity obtained from the best fit considering equal and independent binding sites and the real overall binding affinity ($K_{b}^{I}+K_{b}^{\mathrm{II}}+K_{b}^{\mathrm{III}}$). Plots (**A**,**B**) show the results from the best fitting of Equation (3), while plots (**C**,**D**) consider the simplified Equation (5). In the left plots the number of binding sites was an adjustable parameter, while it was fixed at *n* = 1 in the right plots. The total concentration of ligand was 1 µM in the blue symbols, lower in the green symbols (down to 1 nM), and 10 µM in the red symbols. The value of $K_{b}^{I}$ was 10^4^ M^−1^ (squares), 10^6^ M^−1^ (triangles), or 10^8^ M^−1^ (circles), with lower affinities for the other two binding sites (hollow symbols) or equal affinity for all binding sites (filled symbols).

**Figure 13 ijms-23-09757-f013:**
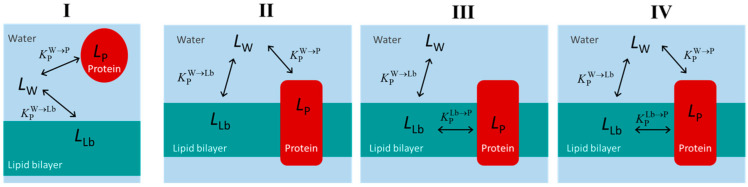
Distribution of ligands in media containing proteins and lipid membranes. Case (**I**)—proteins soluble in the aqueous medium; Cases (**II**–**IV**)—membrane proteins, in the case of ligand equilibration with both binding systems from the aqueous medium (**II**), interaction with the protein from the membrane only (**III**) or from both the aqueous medium and the membrane (**IV**).

**Figure 14 ijms-23-09757-f014:**
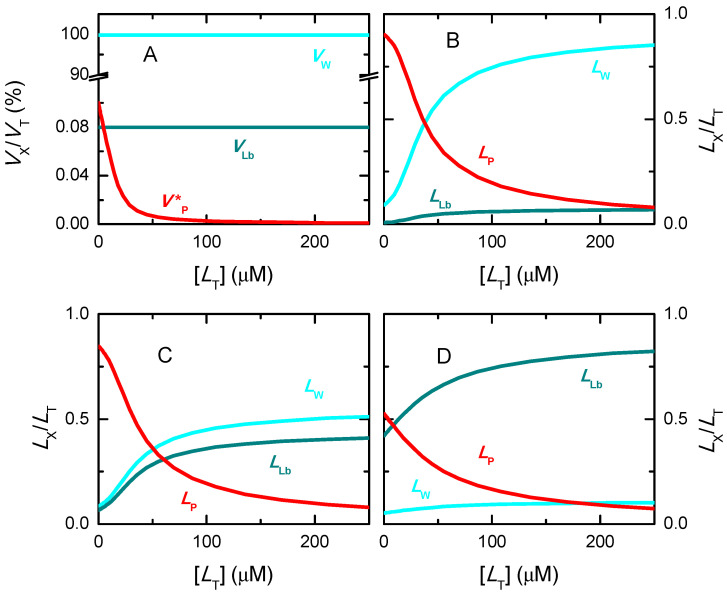
Distribution of a ligand between the aqueous medium, a lipid membrane and a protein soluble in the aqueous medium. The protein is at a concentration of 10 µM and has a molar volume of 50 dm^3^/mol with a single binding site, while a lipid concentration of 1 mM is considered with a lipid molar volume of 0.8 dm^3^/mol. The partition coefficient between the aqueous medium and the protein is the same in all plots $\left(K_{P}^{W\to P}=10^{4}\right)$ while the affinity for the membrane is low in plots (**A**) and (**B**) $\left(K_{P}^{W\to \mathrm{Lb}}=10^{2}\right)$, and increases for plots (**C**) $\left(K_{P}^{W\to \mathrm{Lb}}=10^{3}\right)$ and (**D**) $\left(K_{P}^{W\to \mathrm{Lb}}=10^{4}\right)$.

**Figure 15 ijms-23-09757-f015:**
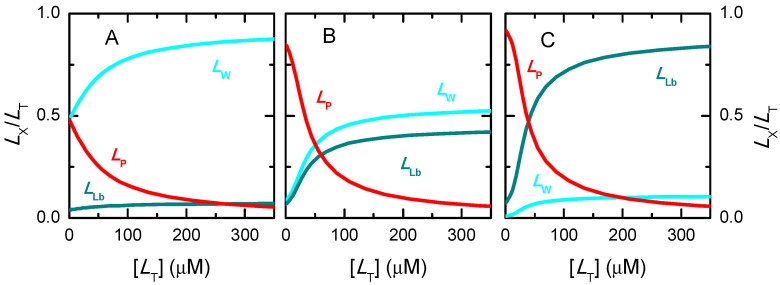
Distribution of a ligand between the aqueous media, the lipid bilayer and the protein in a membrane composed of a lipid bilayer with embedded membrane proteins. The protein and lipid concentrations are the same as in Figure 14, leading to 56% of the membrane being occupied by protein and 44% by the lipid bilayer. The partition coefficient between the lipid bilayer and the protein is the same in all plots ($K_{P}^{\mathrm{Lb}\to P}=10^{2}$) while the affinity for the membrane is low in plot (**A**) ($K_{P}^{W\to \mathrm{Lb}}=10^{2}$ ), and increases for plots (**B**) ($K_{P}^{W\to \mathrm{Lb}}=10^{3}$ ) and (**C**) ($K_{P}^{W\to \mathrm{Lb}}=10^{4}$ ).

## Data Availability

Not applicable.

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
