# Peer review of "Analysis of the Equilibrium Distribution of Ligands in Heterogeneous Media–Approaches and Pitfalls"

_ijms, 2022, doi:10.3390/ijms23179757_

Round 1
Reviewer 1 Report
The manuscript written by Moreno et al. provides a comprehensive and well-organised review of methods commonly used to analyse the equilibrium distribution of ligands in various heterogeneous media such as protein, lipid, and protein-lipid medium. The drawbacks of each method were also analysed, ergo allowing readers to have a better understanding of approaches/formalisms that will be best suited for their system of interest. Hence, I would like to recommend the paper for publication after some minor revisions pertaining to their English writing.
Please rephrase the two sentences in Introduction (line 112-117) for clarity.
It is highly advisable that the authors proofread their paper once again, paying close attention to the revision of awkward sentence structure, to further enhance the quality and clarity of the paper.
Author Response
The manuscript has been revised to further enhance its clarity.
Reviewer 2 Report
The authors of a review manuscript as the tittle “Analysis of the Equilibrium Distribution of Ligands in Heterogeneous Media – Approaches and Pitfalls” describe a continuation of several approaches and formalisms for the analysis of the equilibrium distribution of ligands in the presence of proteins, lipid membranes, or both. Special attention is given to common pitfalls in the analysis, with the establishment of the validity limits for the distinct approaches. Due to its widespread use, the characterization of ligand binding through the analysis of Stern-Volmer plots of protein fluorescence quenching is focused on detail. Systems of increasing complexity are considered, from proteins with single to multiple binding sites, from ligands interacting with proteins only to biomembranes containing lipid bilayers and membrane proteins. A new formalism is proposed where ligand binding is treated as a partition while considering saturation of protein binding sites. This formalism is particularly useful for the characterization of interaction with membrane proteins. I find this study perfectly suitable for publication in International Journal of Molecular Sciences.
Before publication, the authors should address the following minor technical points:
1. p8, line 284: “Bi, LBi Kib “ should have the same nomenclature as in Scheme 8, BI, LBI KIb
2. p8, line 303: “Kbi “ should have the same nomenclature as KIb
Author Response
The "i" in the text accompanying equation (8) represents the distinct binding sites (I, II, or III), this is now clearly indicated in the revised manuscript.